# Brain-controlled modulation of spinal circuits improves recovery from spinal cord injury

Marco Bonizzato [1], Galyna Pidpruzhnykova [2], Jack DiGiovanna[1], Polina Shkorbatova[2,3], Natalia Pavlova[2,3], Silvestro Micera[1,4] & Grégoire Courtine [2,5]

The delivery of brain-controlled neuromodulation therapies during motor rehabilitation may augment recovery from neurological disorders. To test this hypothesis, we conceived a brain-controlled neuromodulation therapy that combines the technical and practical features necessary to be deployed daily during gait rehabilitation. Rats received a severe spinal cord contusion that led to leg paralysis. We engineered a proportional brain–spine interface whereby cortical ensemble activity constantly determines the amplitude of spinal cord stimulation protocols promoting leg flexion during swing. After minimal calibration time and without prior training, this neural bypass enables paralyzed rats to walk overground and adjust foot clearance in order to climb a staircase. Compared to continuous spinal cord stimulation, brain-controlled stimulation accelerates and enhances the long-term recovery of locomotion. These results demonstrate the relevance of brain-controlled neuromodulation therapies to augment recovery from motor disorders, establishing important proofs-of-concept that warrant clinical studies.

[1] Bertarelli Foundation Chair in Translational Neuroengineering, Center for Neuroprosthetics and Institute of Bioengineering, School of Bioengineering, EPFL, Geneva CH-1202, Switzerland. [2] Center for Neuroprosthetics and Brain Mind Institute, School of Life Sciences, Swiss Federal Institute of Technology (EPFL), Geneva CH-1202, Switzerland. [3] Laboratory of neuromorphology, Pavlov Institute of Physiology, St. Petersburg 199034, Russia. [4] The BioRobotics Institute, Scuola Superiore Sant'Anna, Pisa 56025, Italy. [5] Department of Neurosurgery, University Hospital of Lausanne (CHUV), Lausanne CH-1011, Switzerland. These authors contributed equally: Silvestro Micera, Grégoire Courtine.  Correspondence and requests for materials should be addressed to G.C. (email: gregoire.courtine@epfl.ch)

**V**arious neurological disorders compromise the communication between the brain and spinal circuits that produce movement, leading to severe motor deficits. Neural bypasses have restored this communication[1–5]. For example, brain-controlled neuromuscular stimulation reestablished grasping movements in individuals with tetraplegia[1,2]. Similarly, we showed that a direct cortical control over the location and timing of electrical spinal cord stimulation enabled nonhuman primates (NHPs) to perform locomotor movements with a paralyzed leg[3].

While these neural bypasses aim at restoring lost motor functions, there is mounting evidence that their long-term use during rehabilitation may augment functional recovery[6–8]. Our objective was to evaluate this possibility. Specifically, we previously showed[9,10] that gravity-assisted gait rehabilitation enabled by continuous electrical spinal cord stimulation restores voluntary control of locomotion after a severe spinal cord injury (SCI) leading to paralysis. Here, we hypothesized that, compared to continuous stimulation, a direct cortical control over adaptive stimulation protocols during rehabilitation would enhance this locomotor recovery.

Addressing this hypothesis involves a series of engineering challenges. First, clinically relevant settings require a brain–spine interface (BSI) that does not require training prior to injury. Second, to support daily rehabilitation the time required for calibrating the BSI must be minimal. Third, due to the importance of task-specific rehabilitation, the BSI must enable training in varying tasks such as overground walking and stair climbing, not only during automated stepping on a treadmill[10]. To tackle these challenges, we took advantage of our previous developments, both the advanced spinal cord stimulation protocols elaborated in rodents[11–13] and the brain–computer interface technologies conceived in NHPs[3].

We showed that the delivery of epidural electrical stimulations over specific spinal cord locations with a precise timing modulates the degree of leg extension and flexion in rats with severe SCI[11–13]. However, the rats have no control over the stimulation. Consequently, the modulation of leg movements remains involuntary—preventing the use of these stimulation protocols to encourage voluntary locomotion during rehabilitation. Inversely, we reported that NHPs could operate a BSI to execute basic locomotor movements of a paralyzed leg, but they were not able to modulate stimulation protocols to adapt these movements to specific tasks[3]. Due to these limitations, the ability of rehabilitation enabled by a BSI to augment recovery from SCI has not been evaluated.

To remedy these limitations, we developed a BSI that linked cortical activity to spinal cord stimulation protocols. We show that brain-controlled stimulation not only enables the voluntary execution of overground walking and staircase climbing, but also accelerates and improves recovery compared to continuous stimulation when delivered during rehabilitation. These results provide a proof-of-concept on the relevance of brain-controlled neuromodulation therapies to enhance recovery from neurological disorders.

## Results

**Technological platform to implement a BSI**. We aimed at restoring the communication across a severe SCI using a BSI that directly links cortical activity to the modulation of epidural electrical stimulation applied to the lumbar spinal cord during gait rehabilitation. Due to the importance of a rapid link between neural recordings and stimulation protocols, we developed a real-time control system that reads multiunit activity MUA, decodes gait events, configures stimulation parameters and triggers stimulation within iteration loops remaining below 5 ms (Fig. 1).

**Spinal cord stimulation enables locomotion**. Five rats were implanted with a 32-element microwire array into the leg region of the right motor cortex to record MUA[14]. To stimulate the spinal cord electrically, we implanted chronic electrodes over lumbar (L2) and sacral (S1) segments. We also monitored muscle responses elicited by this stimulation using bipolar electromyographic (EMG) electrodes chronically inserted into the left tibialis anterior (Fig. 1). In the same surgery, the rats received a severe contusion SCI using a robotically controlled impact onto mid-thoracic segments.

To evaluate the rats without confounding contribution of the intact forelimbs, we positioned them bipedally in a robotic interface that provides a gravity-assist optimized for each rat[10,15]. At 10 days post-injury, all rats showed complete paralysis of both legs (Fig. 2a). During these evaluations, we did not detect relevant modulations of motor cortex population responses (Fig. 2a).

Serotonin agonists[10] and continuous stimulation applied through L2 and S1 electrodes (40 Hz, 0.1–0.4 mA, 0.3 ms) immediately enabled automated locomotion on a treadmill. However, a systematic dragging of the paws occurred at the beginning of the swing phase (dragging: $18.2 \pm 5.3\%$ of swing duration, $n = 5$, mean $\pm$ SEM). Under these conditions, the spiking activity recorded from the motor cortex displayed cyclic modulations that were phase-locked to the automated (involuntary) locomotor movements of the contralateral leg (Fig. 2a). The modulation depth of these responses strongly correlated with the amount of spared spinal cord tissue ($R^2 = 0.87$, Supplementary Fig. 1), suggesting that sensory afferent feedback might be the primary source of modulation of cortical activity.

To explore this possibility, we recorded cortical ensemble modulation in response to a cutaneous stimulation applied to the paw. The strength of the stimulus was adjusted to avoid leg movements. As anticipated, the sensory stimulation led to small, yet reproducible responses in the leg region of the motor cortex ($n = 5$, $P = 0.03$, $t$-test; Supplementary Fig. 2).

**Decoding foot-off events from cortical population responses**. We then asked whether gait events could be decoded from these modulations. We observed that the cumulative firing of cortical ensemble population systematically increased toward the end of stance and peaked during swing (Fig. 2a). We thus sought to decode the onset of the swing phase from these cortical population responses.

We developed a linear classifier based on least-square fit that tracked neural correlates of foot-off events from the cumulative firing of cortical ensemble population (Fig. 3a). All rats displayed an increase in the cumulative firing of cortical ensemble population toward the end of stance, with a peak of activity during mid-swing (Fig. 3b). Consequently, the cortical activity enabled a high accuracy in the decoding of the swing phase in these animals. The online decoder accurately predicted foot-off events in real-time over extended periods of locomotion in all the tested rats ($n = 5$; $90.2 \pm 2.4\%$ correct detections over 120 s; Supplementary Movie 1). The correct rejection rate (true negatives) peaked as high as $92.4 \pm 1.4\%$ (Fig. 2b). This decoding performance was well above the receiver operating characteristic (ROC) curve of the applied decoding rules (i.e., chance level; Fig. 3c).

Locomotion on a treadmill is a highly repetitive task that may inherently lead to high decoding performance without direct relationships with the actual decoded events. To reject this possibility, we compared the intrinsic variability in the timing of foot-off events with the variability of errors in foot-off detections. We found that the variability of errors in foot-off detections was much smaller than the intrinsic variability of actual foot-off events across all the recorded gait cycles (Fig. 3c).

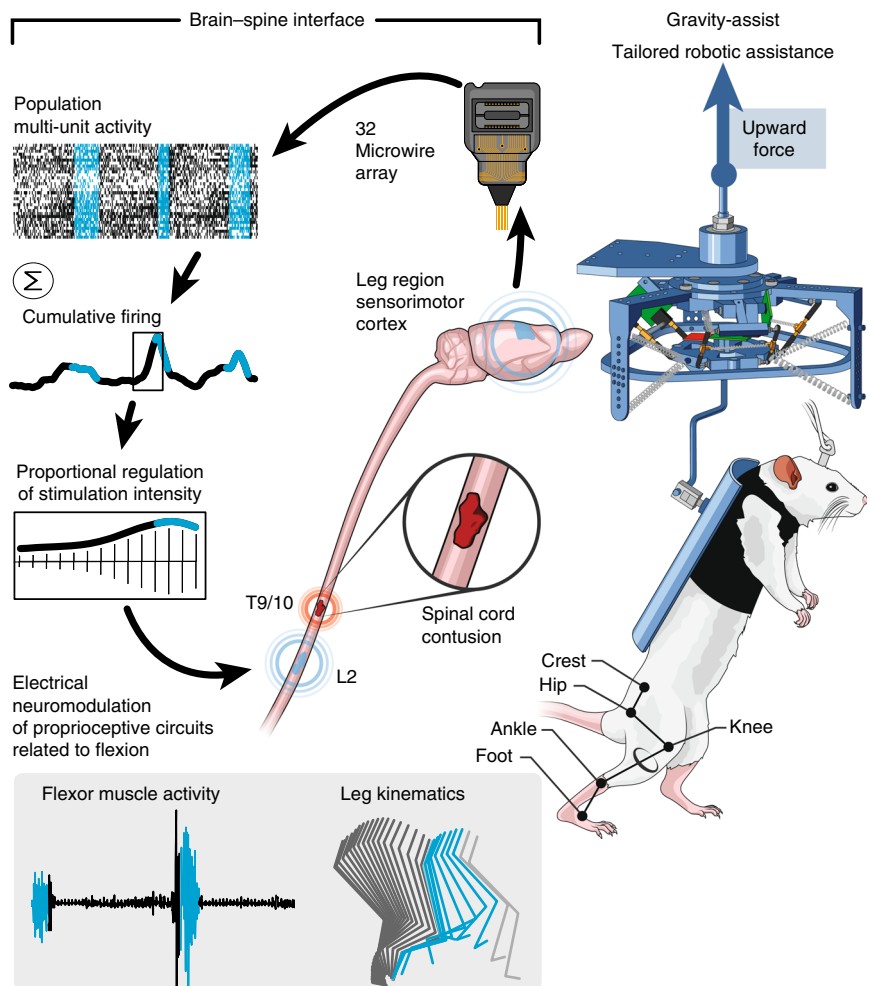

**Fig. 1** Conceptual and technological design of the BSI. The rats were implanted with a microwire array (32 wires) into the leg area of the right motor cortex. The raster plot shows neural recordings over three successive gait cycles. Each line represents spiking events identified from one electrode, while the horizontal axis indicates time. Stance and swing are colored in black and blue, respectively. Two types of BSI were tested. First, a decoder anticipated the onset of the swing phase, which triggered the delivery of stimulation protocols applied to the lumbar spinal cord, wherein motoneurons innervating flexor muscles reside. Second, the cumulative firing calculated from multiunit activity was directly linked to the intensity of stimulation protocols delivered to the same location. Shaded region (bottom left): electromyographic activity of a flexor muscle (tibialis anterior) together with a stick diagram decomposition of leg movements during the stance (dark gray) and swing (blue and light gray) phases of gait. The occurrence of the stimulation is highlighted in blue. During testing, the rats walked in a gravity-assist that personalized the amount of upward force for each rat. Copyright Jemère Ruby (2017)

**A BSI enhancing leg flexion during swing**. We next sought to exploit this decoding to engineer a binary (on/off) BSI through which cortical activity would trigger a spinal cord stimulation protocol targeting flexion components. We previously showed epidural electrical stimulation applied over upper lumbar segments primarily modulates muscle synergies related to flexion[11]. We thus linked the detection of imminent foot-off events to the delivery of a stimulation burst (200 ms) through the L2 electrode. The decoder anticipated foot-off events by 100 ms, which enabled delivery of stimulation at the relevant timing to promote flexion.

Rats were tested 2 weeks post-injury during locomotion on a treadmill. The binary BSI reliably triggered the stimulation bursts over the lumbar spinal cord (true positive: $96.2 \pm 2.4\%$, correct rejection: $97.6 \pm 2.4\%$, $n = 5$). During continuous stimulation, the decoder detected foot-off events $28.9 \pm 14.8$ ms prior to the onset of these events. When the BSI triggered the onset of a stimulation burst, we tuned the decoder to anticipate the detections by 100 ms, which actually occurred $102.2 \pm 24.5$ ms before foot-off events.

During continuous stimulation, the amplitude of the current delivered to the lumbar segments was optimal around $185 \pm 38$

μA on average. The delivery of short stimulation bursts allowed us to increase the stimulation amplitude, which reached $262 \pm 117$ μA on average during locomotion with the binary BSI. The occurrence of well-timed bursts of higher amplitude improved locomotor performance. Compared to continuous stimulation, brain-controlled stimulation led to a $27.3 \pm 10.3\%$ increase in the amplitude of the EMG activity recorded from the tibialis anterior muscle ($P = 0.03$, $t$-test; Fig. 2c). This modulation promoted a significant increase in step height ($62.6 \pm 22.0\%$; $P = 0.03$, $t$-test) and speed of foot movement during swing ($68.4 \pm 15.6\%$; $P = 0.03$, $t$-test), which led to a $44.7 \pm 3.4\%$ reduction in foot dragging duration ($P = 0.03$, signed-rank test, Fig. 2c and Supplementary Movie 1).

The same rats were tested after 3 weeks of rehabilitation, when they had recovered the ability to initiate and sustain (voluntary) bipedal locomotion overground with the gravity-assist. Decoding of foot-off events remained highly reliable. On average, $89.2 \pm 3.5\%$ of foot-off events were correctly identified, with a correct rejection rate of $92.2 \pm 3.4\%$ (Supplementary Fig. 3). As observed on the treadmill, brain-controlled stimulation mediated a significant increase in step height ($35.8 \pm 18.3\%$; $P = 0.04$, $t$-test)

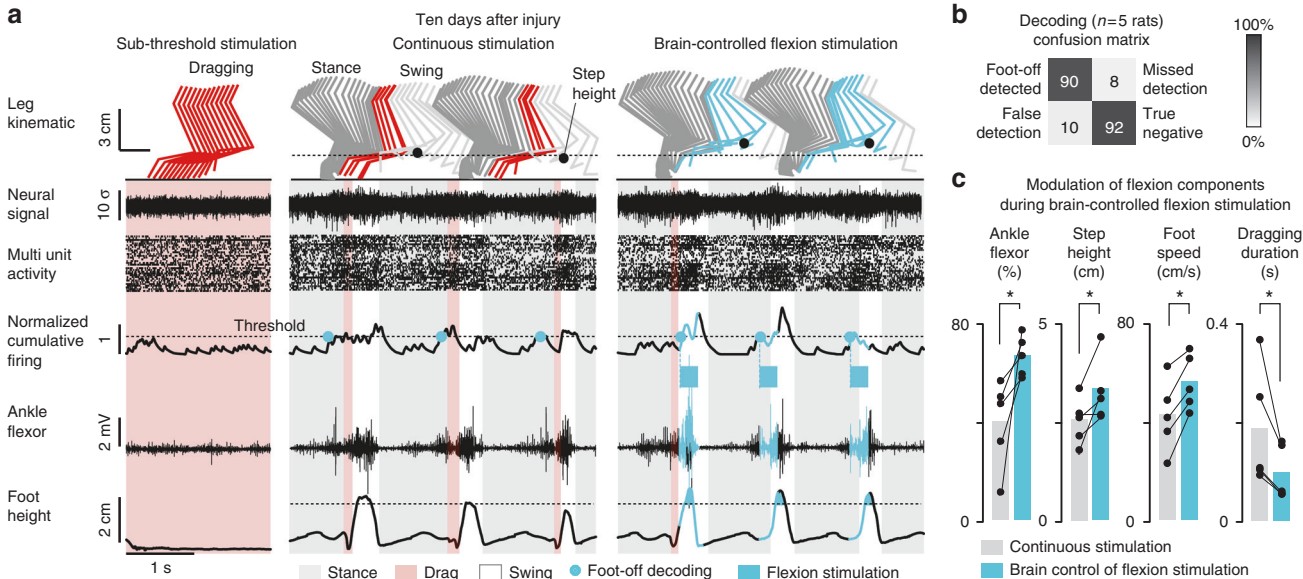

**Fig. 2** Development and validation of a binary BSI after contusion. **a** Recording performed on a treadmill 10 days after the severe contusion. From left to right: sub-threshold stimulation of S1 and L2 segments, stimulation of S1 and L2 segments, stimulation of S1 segment plus brain-controlled of L2 segment (flexion stimulation). From top to bottom: color-coded leg kinematics, neuronal signal from a representative channel, multiunit activity, normalized cumulative firing, electromyographic activity of the tibialis anterior muscle, and vertical displacement of the foot. The gait phases are color coded. The blue dots indicate foot-off events decoded from cortical population ensemble. The region colored in blue highlight the occurrence of stimulation over L2. **b** Confusion matrix of Foot-off decoding calculated across the five rats. **c** Bar plots reporting mean values and individual mean values of parameters modulated during continuous stimulation versus brain-controlled stimulation. The relative activation of the tibialis anterior was calculated as a percent of the maximum activity recorded during locomotion. *, $P < 0.05$

and speed of foot movement during swing ($35.5 \pm 10.4\%$; $P = 0.02$, $t$-test) compared to continuous stimulation ($n = 5$, Supplementary Fig. 3).

**Cortical ensemble population correlates with step height**. The binary BSI reestablished a direct communication between the motor cortex and lumbar spinal cord. While this neural bypass alleviated some of the impairments related to flexion, the amount of transmitted information remained limited, and consequently, gait deficits persisted. Moreover, the rats were not able to adjust the degree of step elevation with the stimulation. Therefore, they could not use this BSI to adapt foot movements to task-specific requirements. We thus sought to increase the resolution of the communication between the brain and spinal cord with the aim to provide a more graded control over foot movements.

To identify additional information embedded in cortical ensemble population, we studied changes in cortical activity in response to rehabilitation. Early after injury, we did not identify correlations between cortical ensemble population and the modulation of gait features such as the step height (0% step height variance explained, $n = 5$, Fig. 4a, b). When rats had regained the ability to produce robust leg movements after training enabled by continuous stimulation, we found a strong linear correlation between the cumulative firing of cortical neurons and the step height. Up to 70% of the variance in step height ($49.9 \pm 6.9$ %, $n = 8$) could be predicted from the cumulative firing rate of cortical ensemble population (Fig. 4b, $P = 4e{-}8$, modified $t$-test, as in Matlab function fitlm). These correlations were consistent with previous findings in healthy rats that associated the peak of cortical activity during swing with the supervision of leg flexion components[14,16–18].

**Graded modulation of flexion components**. We next sought to identify a stimulation strategy that could increase the degree of

leg flexion proportionally to the encoding of step height in cortical activity. We reasoned that the cumulative firing rate of cortical ensemble population could directly control stimulation protocols that adjust the amplitude of the step height throughout locomotion.

We previously showed that epidural electrical stimulation activates motoneurons pre-synaptically through the recruitment of proprioceptive feedback circuits[12]. The activation of these circuits induces stereotyped reflex responses in leg muscles (Fig. 5a).

We exploited this property to map the relationships between the amplitude of stimulation applied to L2 segments and the amplitude of flexor muscle activity. For each rat, we found a functional range over which the stimulation elicited reflex responses restricted to flexor muscles and proportional to the stimulation amplitude ($R^2 = 0.85 \pm 0.05$ within the functional range $\pm 50\%$, $n = 5$, Fig. 5b).

**Proportional BSI enabling the control of step height**. We exploited these results to engineer a proportional BSI that continuously coupled the cumulative firing rates of cortical ensemble population to the amplitude of stimulation applied through the L2 electrode (Fig. 6a).

A new group of five rats participated to these experiments. As early as 7 days after injury, all the rats showed the expected modulation of cortical ensemble population during locomotion (Fig. 6b). Without prior training, the proportional BSI enabled these rats to modulate the amplitude of ankle flexor muscle activity using neural signals directly recorded from the motor cortex (Fig. 6c). Examination of EMG bursts in ankle flexor muscles revealed that muscle activity was elaborated from a succession of reflex responses that were linked to each pulse of stimulation. Since the amplitude of each pulse of stimulation was proportional to the instantaneous cumulating firing of cortical

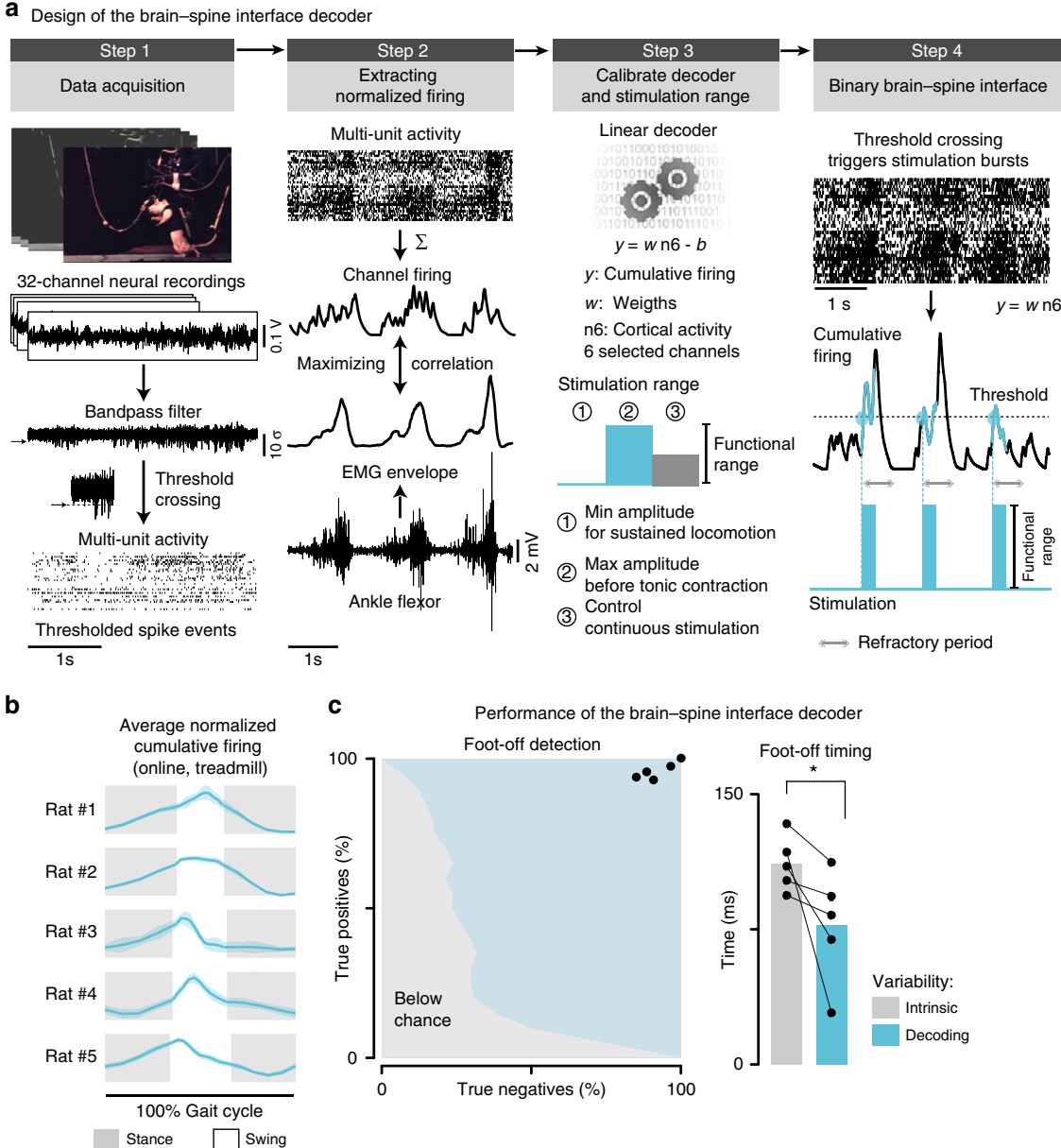

**Fig. 3** Design of the BSI decoder. **a** Successive steps involved in the elaboration of the decoders. Step 1: Neural signals were synchronized with kinematic and muscle activity recordings during locomotion. Each of the 32 channels from the microwire array implanted into the leg area of the motor cortex was filtered, and then transformed in spiking events. The spiking events were calculated from multiunit activity, when passing a threshold that was set manually for each channel. Step 2: The six channels that displayed the largest correlation with the muscle activity measured from the tibialis anterior were isolated. Step 3: A linear decoder linking multiunit activity from the six isolated channels with the control variable (step height) was calibrated for each rat. The linear decoder was then implemented in the online processing pipeline. Step 4: The neural recordings were processed online to obtain spike-rate estimates before passing the resulting cumulative firing through the decoder that tracked neural correlates of foot-off events. When the cumulative firing crossed a threshold corresponding to 100 ms before the predicted occurrence of foot-off events, the pulse generator delivered a 200 ms burst of stimulation over the L2 segment. **b** Online instantaneous firing-rate, averaged across gait cycles, as applied on Step 4 to control the brain–spine interface ($n = 5$ rats on treadmill). **c** Receiver operating characteristic (ROC) illustrating the accuracy of foot-off event detections, which lied well above chance level for all the rats ($n = 5$ rats). Bar plot reporting the variability in the timing of actual and decoded foot-off events across a period of 2 min of continuous locomotion. The intrinsic variability of foot-off events was larger than the average error in foot-off event detections

ensemble population, the amplitude of muscle activity was linearly correlated with motor cortex activity (Fig. 6d, e).

We therefore evaluated whether the proportional BSI restored the relationship between motor cortex activity and step height that otherwise only emerges after several weeks of rehabilitation. As expected, early after injury there was no correlation between the cortical ensemble population and the step height when rats stepped automatically with continuous stimulation ($R^2 = 0\%$ of explained variance; Fig. 6f). The forced link between cortical ensemble population and flexor motoneuron activity re-established this relationship (Fig. 6f). At 10 days post-injury, cortical ensemble activity at foot-off determined $42.2 \pm 4.7\%$ of the variance in step height during the subsequent step ($P = 0.03$, signed-rank test, Fig. 6g), which was comparable to the values measured in rats after 5 weeks of training enabled by continuous stimulation ($R^2 = 49.9 \pm 6.9\%$; Fig. 6g).

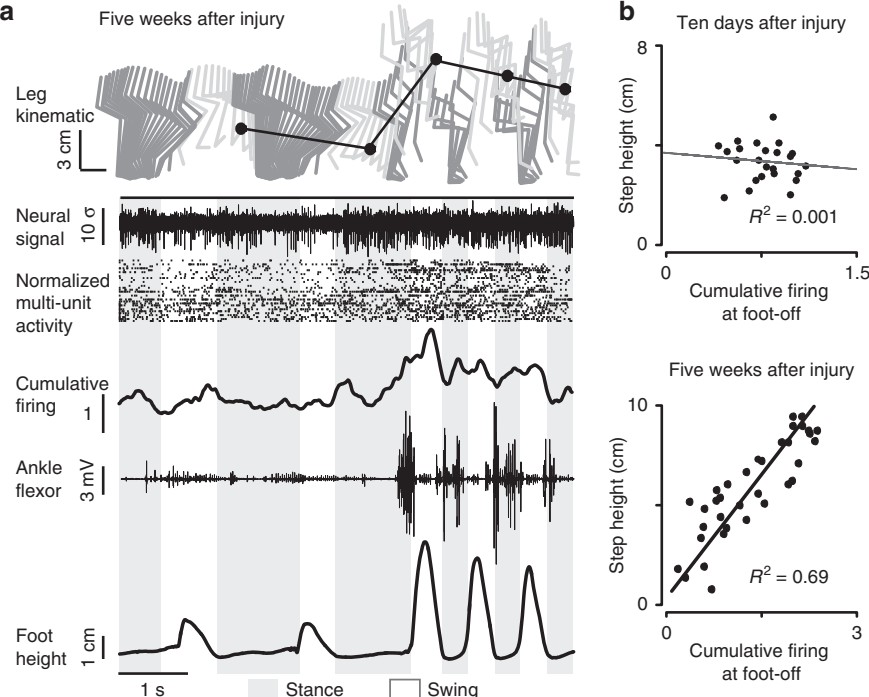

**Fig. 4** Rehabilitation leads to encoding of leg flexion in the motor cortex. **a** Bipedal locomotion recorded on treadmill during continuous stimulation after 5 weeks of gravity-assisted gait rehabilitation. A reward was presented in front of the rat to encourage the variations of foot trajectories. Conventions are the same as in Fig. 2. **b** Correlation between cumulative firing at foot-off and the subsequent step height obtained at 10 days post-injury and after gait rehabilitation. Data are from the rat shown in **a**

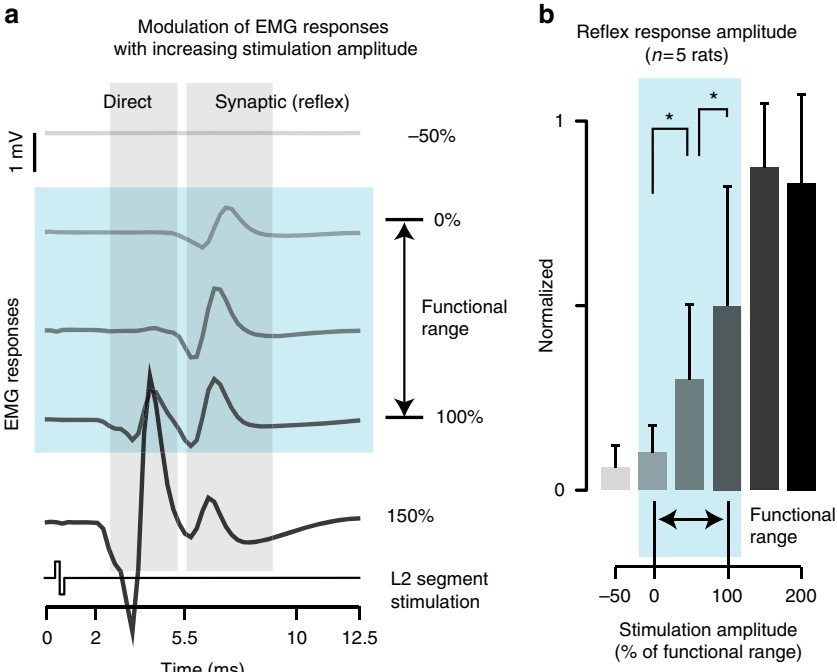

**Fig. 5** Modulation of reflex responses with increasing stimulation amplitude. **a** EMG responses recorded from the tibialis anterior muscles following single pulses of stimulation delivered at L2 with increasing amplitudes of stimulation. The value 0% correspond to the smaller amplitude that was functional to facilitate locomotion. Each response is an average of 10 repetitions. The shaded areas distinguish direct responses (direct stimulation of the motor nerve) from post-synaptic responses (reflex responses), which are elicited from the recruitment of proprioceptive feedback circuits. These temporal windows are defined from the expected latencies and durations of these responses. The blue region highlights the range of amplitudes over which the reflex responses remained functional, i.e. the stimulation evoked a functional increase in leg flexion components during locomotion without causing co-contraction with other muscles. **b** Bar plot reporting the mean amplitude of reflex responses over the entire range of tested amplitudes (mean ± SEM, $n = 5$ rats). The amplitude of these responses was calculated as the integral of the averaged and rectified signals over the temporal window compatible with trans-synaptic responses ($n = 10$ repetitions per amplitude). *, $P < 0.05$

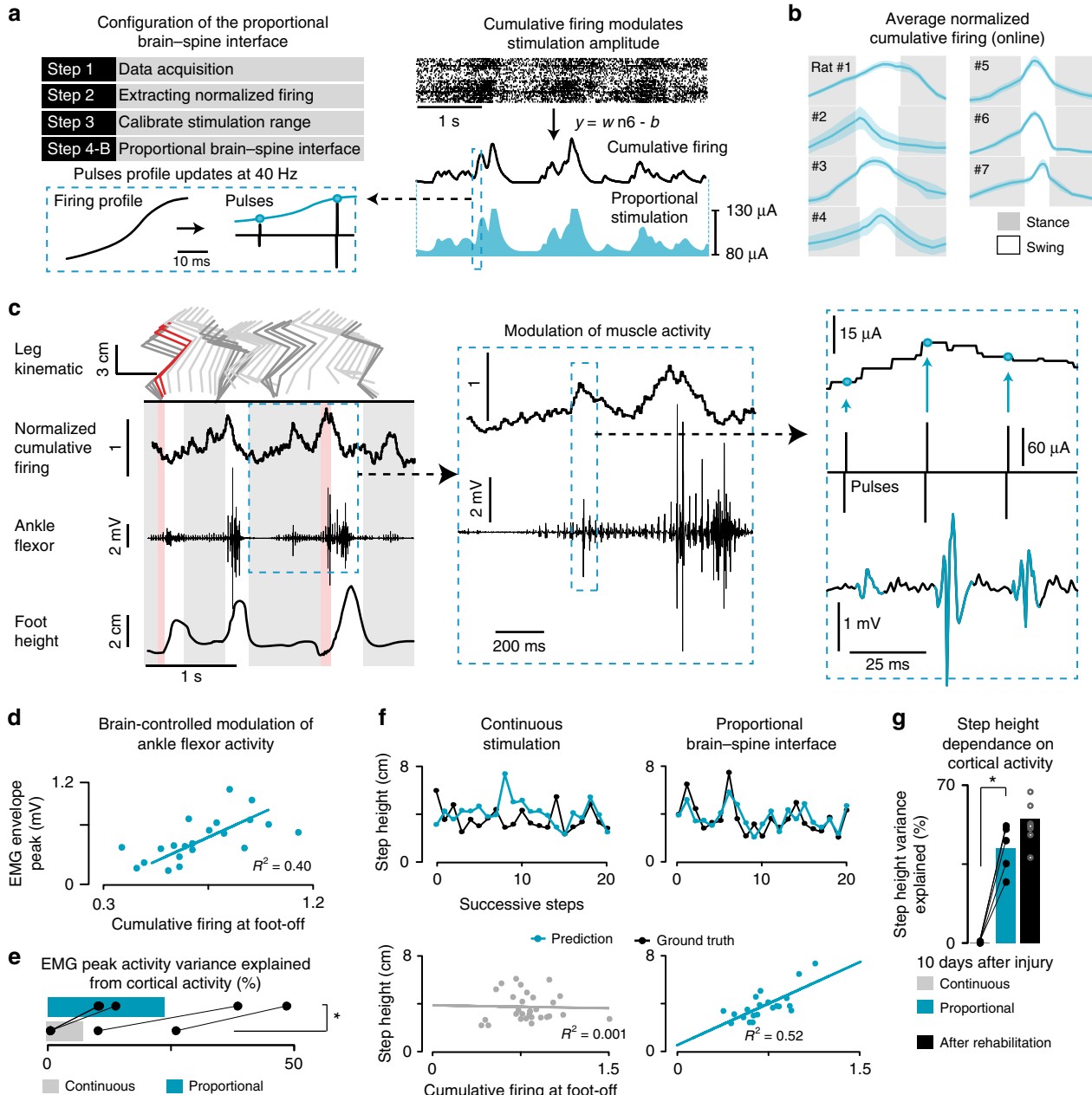

**Fig. 6** Validation of the proportional BSI. **a** Design of the BSI decoder, following the conventions of Fig. 3. Steps 1–3 remain unchanged. Step 4B: The neural recordings were processed online to obtain spike–rate estimates before passing the resulting cumulative firing through the proportional decoder. Every pulse of stimulation (40 Hz) delivered over the L2 segment was instantly shaped to hold an amplitude linearly proportional to the current cumulative firing, as displayed in the inset. **b** Online instantaneous firing-rate, averaged across gait cycles, as applied in Step 4, to control the brain–spine interface ($n = 7$ rats overground). **c** Bipedal locomotion recorded on a treadmill during proportional stimulation 10 days after injury. A reward was presented in front of the rat to encourage the variations of foot trajectories. Conventions are the same as in Fig. 2. First inset: close up of a single gait cycle, illustrating co-variation between ankle flexor EMG magnitude and cumulative firing. Second inset: further zoom-in, showing that the cumulative firing rate (top) set the stimulation amplitude (middle), which induces motor responses in the ankle flexor that is proportional to this amplitude (bottom). **d–e** Peak ankle flexor activity during swing co-variates with cumulative firing at foot-off when the proportional brain–spine interface is active ($n = 5$). **f** Actual step heights (black) and predicted step heights (blue) during a continuous sequence of steps with continuous stimulation and the proportional BSI. The same data are shown in the correlation plots. **g** Bar plots reporting the percent of explained variance in step height from the cumulative firing of cortical ensemble population. The same rats ($n = 5$) were tested with the three conditions of stimulation at 10 days after injury. The same analysis was performed in a group of eight rats that underwent gait rehabilitation with robotic assistance and continuous stimulation for 5 weeks. *, $P < 0.05$

These results show that shortly after injury and without any training, the proportional BSI established a contingency between leg flexor muscles activity and cortical ensemble firing that determined the height of each step during locomotion of otherwise paralyzed rats.

**The proportional BSI improves locomotion**. We then asked whether the proportional BSI improved locomotor performance compared to both continuous stimulation and binary control of stimulation.

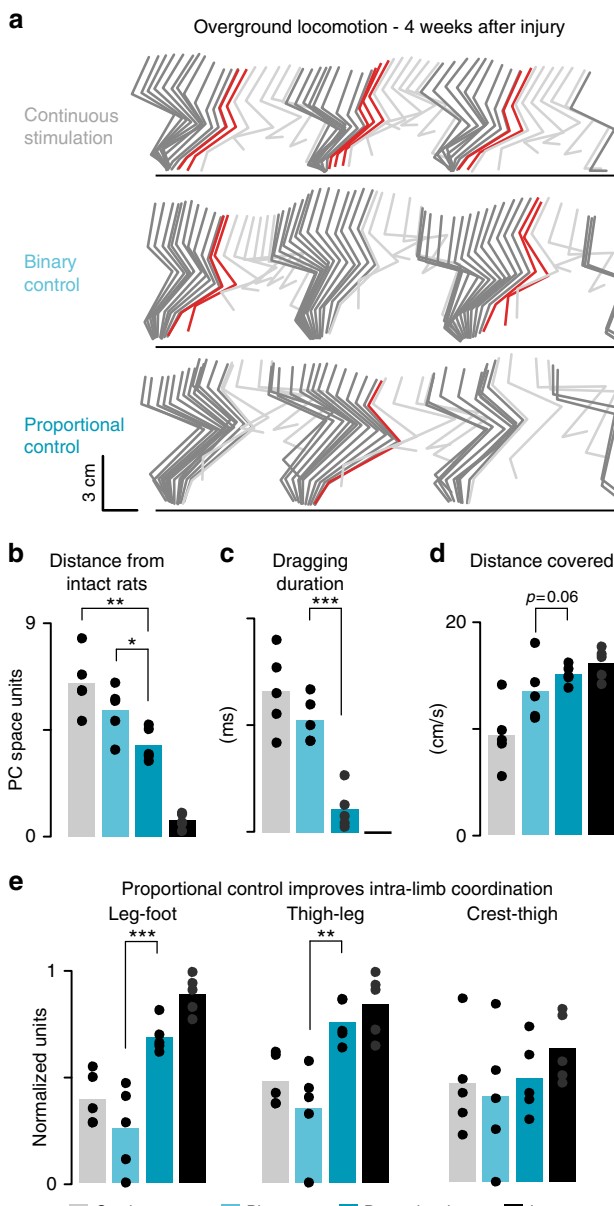

**Fig. 7** The proportional BSI improves overground walking. **a** Stick diagram decomposition of leg movements during a trial with continuous stimulation, binary stimulation and proportional stimulation. Data recorded four weeks after injury. **b** Bar plot reporting the mean values of distances between intact rats ($n = 5$ rats) and the different groups of injured rats in the PC space. This value decreases with improved locomotor performance. **c** Bar plots reporting the mean duration of paw dragging (**d**) and distance covered per unit of time during locomotion. **e** Bar plots reporting the mean values of intra-limb coordination for each joint of the leg, calculated from the cross-correlation values between the oscillations of adjacent segments with respect to the direction of gravity. The abrupt burst of stimulation delivered with the binary control partly disrupted intra-limb coordination. *, $P < 0.05$. **, $P < 0.01$. ***, $P < 0.001$

Leg kinematics was recorded during overground locomotion (Fig. 7a). We applied a principal component (PC) analysis[11] to a large number of parameters calculated from these recordings ($n = 55$ parameters). Locomotor performance was quantified as the distance from intact rats in the space defined by the first three PCs (explained variance, 63.0%).

Compared to both continuous stimulation and binary control of stimulation, the proportional BSI enabled rats to produce gait patterns that resembled more closely those recorded in intact rats ($P = 0.002$ and $P = 0.02$ respectively, $t$-test, $n = 5$, Fig. 7b). In particular, the proportional control over stimulation protocols significantly reduced the amount of paw dragging ($P < 0.001$, $t$-test, Fig. 7c), increased the distance covered overground per unit of time ($P = 0.06$, $t$-test, Fig. 7d) and improved the coordination between the oscillations of lower limb segments (cross-correlation between leg-foot $P < 0.001$, thigh-leg $P = 0.003$, crest-thigh $P = 0.3$, $t$-test, Fig. 7e). Instead, the sudden burst of stimulation triggered by the binary controller tended to disrupt intra-limb coordination (Fig. 7e).

These analyses demonstrate the physiological and functional superiority of the proportional BSI to enable locomotion after a severe SCI.

**The proportional BSI enables volitional modulation of gait.** We next evaluated whether the rats were able to exploit the proportional BSI to adjust leg movements to task-specific requirements. For this purpose, we tested the rats 1 month after injury during a stair climbing task that required a voluntary increase in foot clearance during swing (Fig. 8a).

When approaching the staircase, all rats ($n = 5$) increased the cumulative firing of cortical ensemble population ($P = 0.008$, $t$-test; Fig. 8b, e). The resulting increase in stimulation amplitude produced a proportional augmentation of the subsequent step height that allowed the rats to pass the staircase successfully (Fig. 8d). During continuous stimulation, the same rats failed to produce the same increase in step height ($P = 0.05$, $t$-test; Fig. 8c), which resulted in a significantly larger number of falls and tumbles (foot hitting the staircase) compared to executions with the proportional BSI ($P = 0.04$, $t$-test; Fig. 8d).

These results showed that the increased amount of information transmitted across the injury through the proportional BSI enabled otherwise paralyzed rats to regain some volitional control over foot clearance in order to climb up a staircase. These results suggest that rats were able to anticipate the modulation of motor cortex activity to mediate a functional increase in leg flexion.

**Gait modification is contingent on the brain controller.** We then studied the contingency on the activity of cortical neurons for the delivery of proportional stimulation protocols that enable voluntary adjustments of leg movements. Specifically, we asked whether more accessible physiological signals could be harnessed to detect foot-off events and modulate stimulation protocols to achieve a desired step height.

For this purpose, we selected the EMG activity of the ankle flexor muscle. As early as 10 days after injury, we found that ankle muscle activity enabled decoding foot-off events with a high degree of accuracy, which was comparable to the levels obtained with cortical activity (Fig. 9a). This result suggests that a controller based on ankle muscle activity may be sufficient to operate a binary controller.

We then studied the possibility to decode the step height from ankle muscle activity. While the increase in the cumulative firing of cortical ensemble population anticipated the vertical displacement of the foot, the EMG burst was logically concomitant to foot movements (Fig. 9b). Consequently, the amplitude of the subsequent step height could not be decoded from the activity of ankle muscles at foot-off (Fig. 9c). Moreover, during locomotion enabled by continuous stimulation, ankle muscle activity failed to predict the cumulative firing of cortical ensemble population, both early and late after the injury (Fig. 9d).

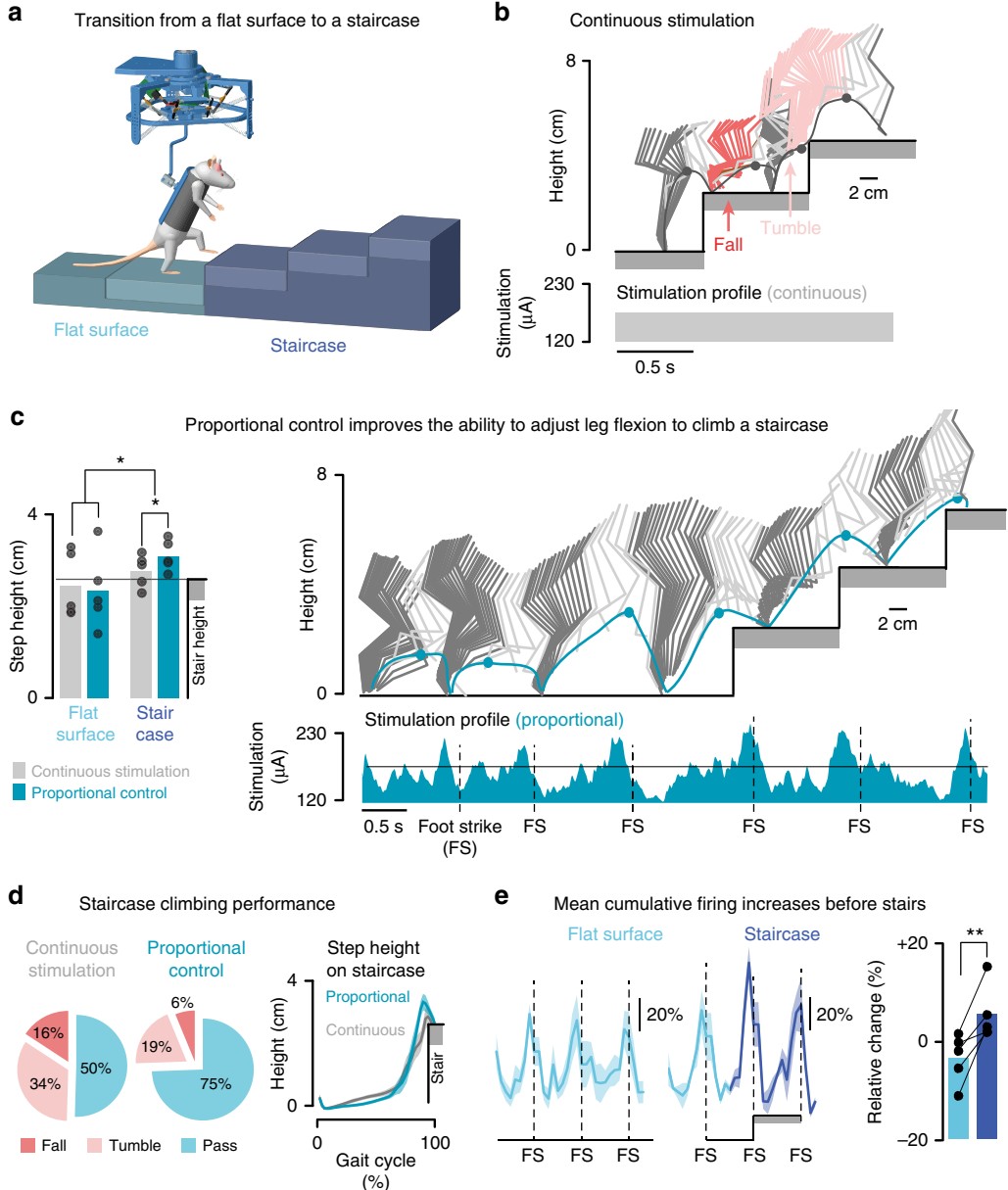

**Fig. 8** The proportional BSI improves stair climbing. **a** Scheme of the staircase climbing task. **b** Stick diagram decomposition of leg movements and foot trajectory during a trial with continuous stimulation. The instantaneous stimulation amplitude is indicated at the bottom. **c** Bar diagram representing the average step height in trials on a flat surface or when a staircase is positioned in front of the animal, with continuous stimulation and proportional control ($n = 5$). Right: Stick diagram decomposition of leg movements and foot trajectory during a trial with proportional brain–spine interface. The instantaneous stimulation amplitude is indicated at the bottom. **d** Circular plots reporting the relative percent of trials with a successful step onto the elevated platform (pass), a tumble (hitting the foot against the staircase) and a fall when climbing the staircase with continuous stimulation or proportional brain–spine interface ($n = 5$ rats). Quantification was performed blindly by two independent experts and averaged. Right: Mean trajectory of the foot ($n = 10$ trials) when passing the first staircase with continuous and with proportional brain–spine interface ($n = 5$ rats). **e** Continuous cumulative firing when progressing on the flat surface and during the first two steps on the staircase. Bar plots reporting the mean change in cumulative firing when progressing along a flat surface and up the staircase ($n = 5$ rats). *, $P < 0.05$. **, $P < 0.01$

We concluded that ankle muscle activity could not be used as a surrogate of cortical activity to enable rats to tune stimulation protocols voluntarily, thus highlighting the importance of the proportional BSI for task-specific adjustments.

**Proportional BSI enhances improvements with rehabilitation.** Finally, we exploited these developments to test our main hypothesis. We evaluated whether rehabilitation enabled by the proportional BSI mediated a superior recovery compared to the same training regimen with continuous stimulation.

Two groups of rats participated in these experiments. The first group ($n = 7$) was trained during 5 weeks, 5 times per week for 30 min with continuous stimulation, while the second group ($n = 6$) was trained the same amount of time with the proportional BSI. Both groups of rats were trained for approximately 10 min on the treadmill before performing locomotion overground and along a staircase during 20 min. We did not add a group of rats trained with the binary BSI since this paradigm was inferior to the proportional BSI to improve locomotion and did not enable task-specific training. Locomotor performance was evaluated weekly

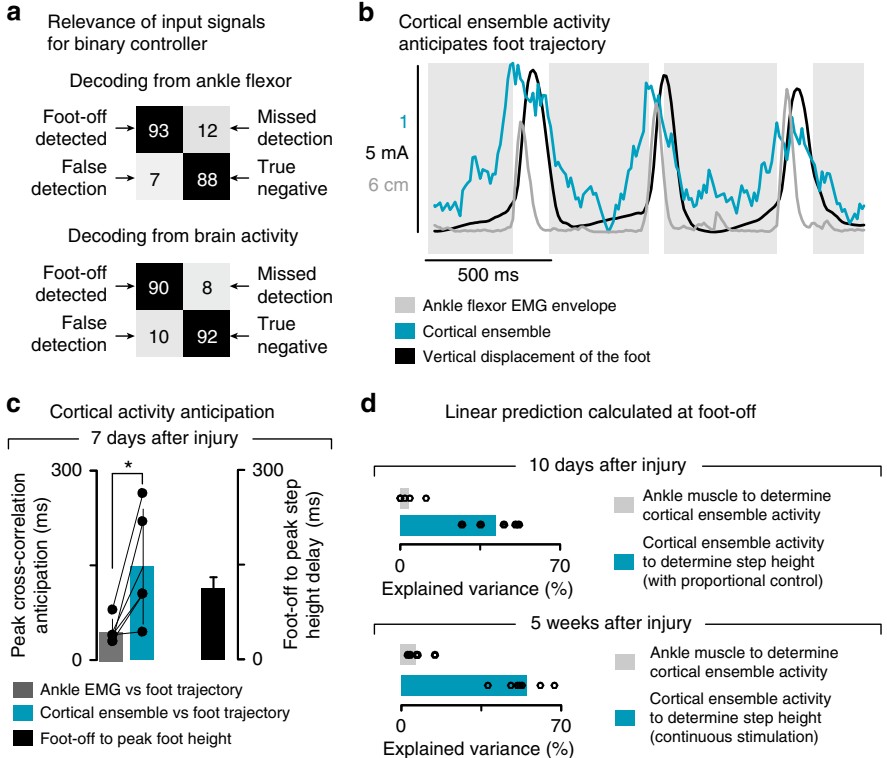

**Fig. 9** Proportional BSI encodes information not found in muscle activity. **a** Binary control: confusion matrix (*n* = 5) of Foot-off decoding calculated online across the rats using cumulative firing as in Fig. 2 (bottom) and ankle flexor EMG envelope (top). **b** Proportional control: cumulative firing, ankle flexor envelope and foot trajectory during three consecutive steps. **c** Bar diagram representing the anticipation of cortical activity to leg trajectory is superior to the length of the flexion phase of swing, while the EMG correlates at shorter latencies (*n* = 5 rats). **d** Ankle flexor muscle envelope evaluated at foot-off fails to explain the final position of the foot during swing 10 days after injury or 5 weeks after injury. Conversely, cortical ensemble activity explains more than 50% of this variance when the stimulation is controlled by the proportional brain–spine interface 10 days after injury, or when tested with continuous stimulation after several weeks of rehabilitation. *, *P* < 0.05

overground with the gravity-assist and continuous stimulation. We quantified locomotor performance using the PC analysis described above (explained variance by PC1-3, 62.0 %; Fig. 10a).

The extent of spinal cord damage was similar across trained rats (tissue sparing, continuous stimulation: 14.8 ± 5.1%, BSI: 12.0 ± 1.0%; Supplementary Fig. 4). During the first 2 weeks post-injury, no difference was detected between both groups. Over the course of training, all rats showed progressive improvements (*P* < 0.0001, *t*-test/ signed-rank test; Fig. 10a, b). However, from the third week and until the end of rehabilitation, rats trained with the proportional BSI exhibited significantly better locomotor performance than rats trained with continuous stimulation (*P* = 0.03, *t*-test; Fig. 10b). Concretely, important gait features such as weight-bearing capacities (*P* = 0.01, signed-rank test; Fig. 10c) and foot speed during swing (*P* = 0.05, *t*-test; Fig. 10c) improved significantly more in response to training enabled by the proportional BSI (Supplementary Movie 2).

Over the entire duration of the rehabilitation program, the daily calibration of the BSI only required a few minutes of fine-tuning and could then be operated with minimal changes throughout the training sessions.

## Discussion

We developed a proportional BSI that established a continuous link between the brain and spinal cord located below a severe, clinically relevant contusion SCI. Brain-controlled stimulation of the denervated spinal cord not only immediately enabled robust movements of the paralyzed legs that supported the execution of complex tasks such as stair climbing, but also improved recovery compared to continuous stimulation when delivered during rehabilitation. We discuss the implications of these results for the development of BSI technologies, speculate on the possible mechanisms through which this paradigm enhanced recovery, and consider the next steps for clinical applications.

In healthy rats, the motor cortex contributes minimally to the production of locomotion[19,20]. The ablation of the motor cortex only leads to transitory impairments in skilled behaviors. However, task-dependent leg movements are robustly encoded in cortical ensemble population[14,17,18,21]. Consequently, these cortical signals can be readily exploited to design brain–computer interfaces for locomotor applications[16,21,22].

We confirmed these findings after a severe SCI. Indeed, we found that the onset of leg flexion could be robustly decoded from cortical ensemble population in rats. In turn, this decoding effectively triggered stimulation protocols that enhanced leg flexion during swing. Moreover, we could maintain a continuous link between the modulation of cortical population responses and the amplitude of stimulation protocols. As early as 10 days after injury, this link allowed paralyzed rats to modulate the activity of the leg musculature in order to produce locomotor movements that resembled those observed in rats trained for several weeks with continuous stimulation[10]. Moreover, the rats exploited the proportional link to mediate a volitional increase in foot clearance in order to climb a staircase. Importantly, these functional improvements did not require training prior to the injury nor a phase of learning after the injury. Indeed, the neural bypass linked cortical activity that naturally occurs during locomotion to the modulation of proprioceptive feedback circuits that are naturally engaged in the production of locomotion[12]. This ecological

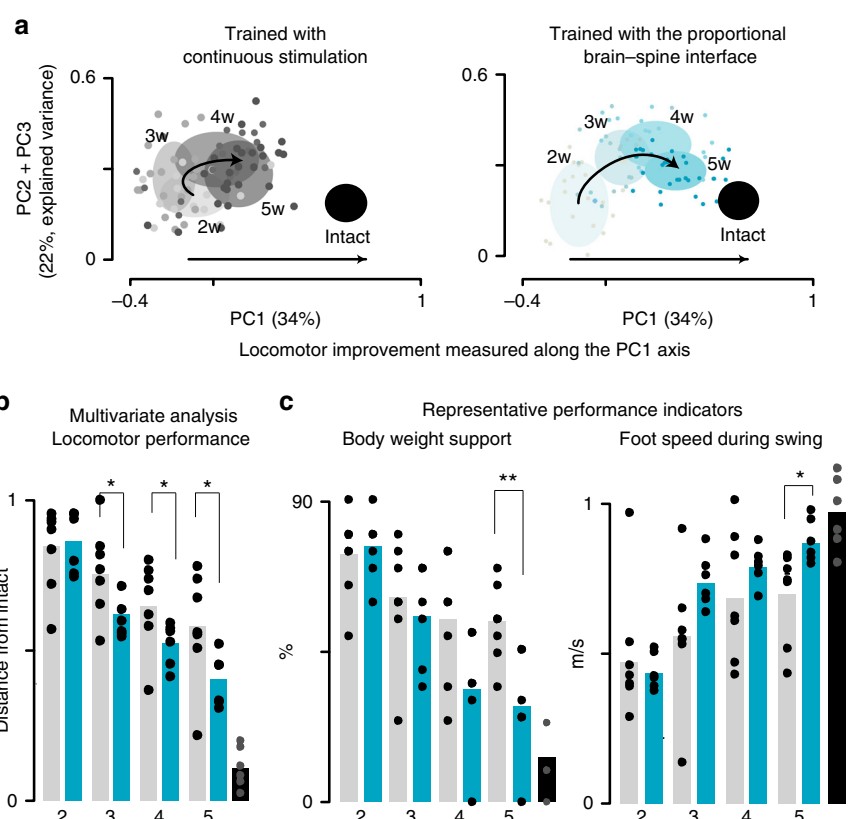

**Fig. 10** Rehabilitation enabled by the proportional BSI improved recovery. **a** Individual gait cycles recorded during overground locomotion with continuous stimulation every week, from week 2 to week 5, are displayed in the PC space for two representative rats. The PC analysis was applied on all the gait cycles from all rats at all the time-points (w, week). **b** Bar plot reporting the mean values of distances between trained rats ($n = 6$ and 7 rats for the trained group) and intact rats ($n = 6$) in the PC space over the course of the gait rehabilitation program. This value decreases with improved locomotor performance. **c** Bar plots reporting the mean values of body weight support capacities and maximum foot speed during swing. *, $P < 0.05$

approach[23] enabled a rapid calibration of the neural bypass, and its immediate use for the production of adaptive locomotor movements and daily gait rehabilitation. These features are essential for the practical deployment of a BSI that can support gait rehabilitation every day in clinical settings.

We previously developed a wireless BSI that triggered spinal cord stimulation protocols inducing extension and flexion movements of a paralyzed leg in NHPs[3]. As observed in rats[11], increasing the amplitude or frequency of stimulation protocols mediated a linear increase in the extent of the extension and flexion of the leg. However, the adjustment of stimulation protocols was pre-programmed. The animals had no control over the parameters of stimulation. Theoretically, the continuous proportional controller developed in rats could translate into phase-dependent control algorithms that continuously modulate the degree of extension and flexion for each leg. Such a state-dependent, proportional BSI has the potential to mediate a markedly more refined prosthetic control of the legs.

The viability of this control strategy is contingent on the similarities between the source of cortical modulation in rodents and primates. As we discuss below, our results suggest that the modulation of cortical activity during stepping on a treadmill was essentially driven by sensory feedback. Instead, NHPs displayed modulation of cortical activity prior to initiating leg movements, suggesting that cortical ensemble population encodes motor intentions. However, we showed that rats also learned to modulate cortical activity to mediate intentional increases in foot clearance to climb a staircase. As in NHPs, this modulation occurred prior to the desired change in leg trajectory, suggesting that this task-specific modulation reflected motor intentions. Finally, two decades of research in brain–computer interface showed that people with long-lasting tetraplegia are capable of modulating leg motor cortex activity without overt movement[24]. They can also control robotic arms[25,26] and neuromuscular stimulators[1,2] to execute manual tasks using only signals recorded from the motor cortex. These results suggest that the relatively simple decoding strategies employed in this study may be part of a solution for humans. Future experiments will have to evaluate the viability of a combined strategy.

The proportional BSI enabled the rats to regain voluntary control over task-specific leg movements but even after rehabilitation, the rats were dependent on the presence of stimulation to locomote. After less severe SCI and/or more extensive training regimen, some recovery can occur without any stimulation[9]. However, the availability of a practical BSI would still be desirable for the more severely affected patients, indicating the importance of developing long-lasting prosthetic technologies for clinical applications. To this respect, recordings of MUA from intracortical neural probes extinguish rapidly, which would prevent the long-term use of a proportional BSI operating with such probes in humans. However, the robustness of the proportional controller came from the relative simplicity of the control signal—a simple

summation of MUA recorded from cortical ensemble population that required minimal daily recalibration. Accordingly, local field potentials, which allowed human participants to control brain–machine interfaces for several years[27], may be sufficient for operating a proportional BSI.

Despite its limited contribution to locomotor control, there is growing evidence that the motor cortex of rats plays a critical role in motor recovery after injury[10,28–30]. Our results are consistent with this model. We observed a progressive recovery of voluntary leg movements in response to rehabilitation that coincided with the emergence of robust correlations between cortical ensemble population and leg flexion components during locomotion. These results suggested that the motor cortex directly contributed to the production of movement following rehabilitation. Indeed, we recently showed that the delivery of continuous spinal cord stimulation during rehabilitation promotes an extensive reorganization of cortical, brainstem and spinal projection circuits that mediates a motor cortex dependent recovery of voluntary leg movements after SCI[9]. We propose that the BSI not only promoted a similar neuroplasticity of residual connections during gait rehabilitation, but also triggered additional use-dependent changes through Hebbian-like plasticity[6,31].

Addressing this hypothesis will require extensive experiments that expand beyond the scope of the present study. To guide these future experiments, we speculate on the mechanisms that may be responsible for the additional functional improvements when gait rehabilitation is performed with a BSI. Evidence suggests that the modulation of motor cortex activity during automated locomotion early after injury was essentially driven by sensory afferent feedback. At this early stage, the electrochemical neuromodulation therapy enables the lumbar spinal cord to interface sensory information with the coordinated recruitment of motor circuits in order to produce locomotion[32]. However, supraspinal centers have not been associated with this execution[10]. Therefore, the observed modulation of cortical ensemble population likely resulted from residual neural inputs arising from spared ascending afferent pathways. Indeed, cutaneous stimulation of the paw elicited reproducible cortical responses in the absence of overt movements, suggesting that sensory pathways mediated the observed modulation of cortical activity[33].

In this scenario, the BSI linked the sensory-driven modulation of cortical ensemble population with stimulation protocols that tuned proprioceptive feedback circuits[12]. In turn, this modulation of afferent pathways was directly fed back to cortical ensemble population. Consequently, the neural bypass established a continuous closed-loop connection between cortical and spinal ensemble populations, thus creating the necessary conditions to reinforce the connection between these two populations through Hebbian-like plasticity[6, 31]. Consistent with this interpretation, we previously showed that the mice lacking proprioceptive feedback circuits failed to recover from a partial SCI, whereas wild type mice regained robust locomotion after the same lesion[34].

A similar interpretation has been invoked to explain an unexpected neurological recovery in response to a brain–computer interface-based gait rehabilitation[35]. Paraplegic individuals were trained in an exoskeleton that was actuated based on non-invasive brain recordings. In addition, artificial sensory feedbacks were delivered to both arms in order to feed leg movement-related information to the spinal cord above the injury. Over time, this closed-loop system restored sensation in some of the originally denervated dermatomes.

We propose that, both in rats and humans, these gait rehabilitation programs closing the loop between circuits located above and below the injury increase use-dependent neuroplasticity of residual connections[8,10,36]. Bidirectional spike-timing-dependent neuroplasticity would be the most probable mechanism steering this reorganization[6–8]. However, this interpretation remains speculative. The physiological, anatomical, and molecular mechanisms that may support or invalidate this potential explanation will require additional studies.

The proportional BSI only targeted flexion components and was tested in rodent models. Despite these limitations, the results provide an important proof-of-concept on the relevance of brain–controlled stimulation of the denervated spinal cord to accelerate and augment recovery from SCI. The ability of the BSI to restore locomotion in a NHP model of transient leg paralysis reinforces this conclusion[3]. For these previous experiments in primates, we developed a wireless BSI integrating intracortical arrays[25,26], wireless modules[37] and pulse generators that have been approved for research applications in humans (https:// clinicaltrials.gov/show/NCT02936453). Recently, we also conceived a gravity-assist algorithm that allows gait rehabilitation of paraplegic individuals in natural conditions[15], as implemented in the present study for rodents. This conceptual and technological framework establishes the appropriate conditions to evaluate the efficacy of BSI-based gait rehabilitation for augmenting locomotor recovery in paraplegic individuals.

## Methods

**Animals**. Experiments were conducted in adult female Lewis rats (between 4 and 7 months old, 200–220 g body weight). All experimental procedures were approved by the Veterinary Office of the Canton Vaud. Each rat was individually housed in a transparent cage with access to food and water ad libitum. The room was kept on a 12 h light/dark cycle at 22 °C ambient temperature. Prior to surgery, all the rats were acclimatized to walk freely along the runway.

**Surgery**. All surgical procedures and post-operative care for rats with SCI followed well-establish procedures[10,11,14]. Aseptically and under general anesthesia, a 32-channel microelectrode array (Tucker-Davis-Technologies, USA) was inserted into layer V of the leg region of the right motor cortex, which we previously identified anatomically and electrophysiologically[7]. Bipolar electrodes were inserted into the left (contralateral) tibialis anterior muscles to record EMG signals. Two wire electrodes were sutured to the dura over the dorsal aspect of lumbar (L2) and sacral (S1) segments to deliver electrical stimulation[10]. Rats received robotically controlled contusion injury delivered at the T9–T10 spinal level using the Infinite Horizon impactor (Precision Systems and Instrumentation, USA). The impact force was set at 250 kdyn. Two rats were excluded from the study since they did not show modulation of the cortical activity during locomotion. Post-mortem evaluation of tissue damage revealed excessive tissue damage in these rats.

**Groups**. Multiple groups of rats participated to these experiments. The list of experimental procedures conducted on the various rats is available in Supplementary Table 1.

**Locomotor training**. Five minutes prior to each training session, rats received an intraperitoneal injection of Quipazine and subcutaneous injection of 8-OH-DPAT[10]. The rats were trained on a treadmill (11 cm s$^{-1}$) and overground in a bipedal posture that encourages volitional control of the legs to walk forward toward a food reward. During training, electrical stimulation was delivered continuously over the L2 and S1 electrodes (monopolar pulses, 40 Hz, 50–350 µA, 0.2 ms). Each training session lasted 30 min and took place 5 day per week, starting from 7 days post-injury.

**Recordings of kinematic and muscle activity**. Well-established procedures for kinematic and muscle activity recordings were used in this study[10,11,14]. Bilateral leg kinematics were captured using the high-speed motion capture system Vicon (12 infrared and 2 digital video cameras, 200 Hz; Vicon, UK). EMG signals were collected (2 kHz, 10–1000 Hz bandpass filtered) using the same system.

**Analysis of kinematic and muscle activity**. A total of 55 parameters quantifying kinematic features were computed for each gait cycle. All the computed parameters[10,11,14] are detailed in Supplementary Table 2. To evaluate differences between experimental groups (rehabilitation), we implemented a statistical procedure based on PC analysis. PC analyses were applied on data from all individual gait cycles for all the rats together. Data were analyzed using the correlation method, which adjusts the mean of the data to 0 and the standard deviation to 1. This method of normalization allows the comparison of variables with disparate values (large vs. small values) as well as different variances[38]. Locomotor

performance was quantified as the Euclidean distance from intact rats in the PC space defined by the first three PCs.

**Cortical recordings**. Intracortical voltage signals were sampled at 24 kHz, pre-amplified, digitalized and filtered online (bandpass, 0.7–3 kHz) using the real-time BioAmp processor from Tucker–Davis Technologies (USA). We calculated MUA from the neural signal crossed a threshold value defined visually for each channel. Spike counts were collected in bins of 10 ms and then smoothed with a finite impulse response (FIR) filter with Gaussian sample decay of 80% in 40 ms.

**Spinal cord stimulation**. Epidural electrical stimulation (Biphasic monopolar, 0.2 ms, 100–400 µA, 40 Hz) was delivered through the chronically implanted electrodes at L2 and S1 segments using the system used for neural recordings (Tucker–Davis Technologies, USA). During experiments with the BSIs, the stimulation delivered over the S1 segment was maintained constant (continuous stimulation).

**Cortical decoding**. For each rat, we identified the six channels with the MUA that correlated most with the envelope of the tibialis anterior muscle. The EMG signal was rectified and low-pass filtered at 10 Hz before applying the Matlab function corrcoeff between this signal and each neural signal. We then built a linear combination of the normalized MUAs from these six channels termed normalized cumulative firing. This variable was used as the control variable for both BSIs. The hard-real-time controller operated within cycles of 12 kHz. The normalized cumulative firing was thus a continuous signal generated with a frequency of 12 kHz. Its value instantly increased by a fixed equal quantity $\delta$ whenever a spike was detected in one of the six selected channels. The normalized cumulative firing decreased with a Gaussian decay over time (FIR filter with sample decay of 80% in 40 ms). The value $\delta$ was normalized in order to ensure that the peak normalized cumulative firing equals one over the duration (approximatively 30 s) of the data used for calibrating the decoder.

**Binary BSI**. Whenever the normalized cumulative firing crossed a manually selected control threshold corresponding to a latency of 100 ms prior to foot-off event, the controller delivered a stimulation burst (200 ms) over the electrode located at L2. A refractory period of 600 ms and 600–800 ms was set before the next event detection on the treadmill and runway, respectively. During locomotion along the runway, an additional silent period of 1.4 s was inserted after the first event detection (initiation) in order to account for the time required for gait initiation. When evaluating decoder accuracy, all the detected foot-off events that occurred within a window of [−200, 100] ms centered on the actual foot-off (roughly 25% of the average step cycle) where considered as true positives (foot-off detected). A missed detection is reported in case no detection occurs within this window. True negatives (correct rejection) are scored for lack of detections in the timespan between two flexion windows, false detections otherwise. The ROC of the decoding rules is computed by Monte Carlo method, as the average performance of the decoding algorithm to a white noise input, for different noise power amplitude (e.g., a flat zero noise input would lead to 100% true negatives and no correct foot-off detection).

**Proportional BSI**. We built a linear relationship between the normalized cumulative firing and the amplitude of stimulation delivered to the L2 segment. The currents were constrained with a range of functional values that were identified based on behavioral recordings. Specifically, the lower and upper bounders were defined as the lower and higher current amplitudes capable of mediating stable locomotion on a treadmill while avoiding dragging (lower) or co-contraction of antagonist muscles (upper), respectively. This tuning required 2–5 min prior to each experimental session, although these thresholds remain globally stable over time. All the parameters were kept constant across the recording sessions.

**Immunohistochemistry and neuromorphological evaluation**. Rats were deeply anesthetized using an i.p. injection of 0.5 ml Pentobarbital-Na (50 mg per mL) and transcardially perfused with approximately 80 ml Ringer's solution containing 100 kIU per L heparin (Liquemin, Roche, Switzerland) and 0.25% NaNO$_2$ followed by 300 ml of cold 4% phosphate buffered paraformaldehyde, pH 7.4 containing 5% sucrose. The brain and spinal cord were removed, post-fixed overnight, and later transferred to 30% sucrose in phosphate buffer (PB) for cryoprotection. After 3 days, the tissue was embedded in Tissue Tek O.C.T (Sakura Finetek Europe B.V., The Netherlands), frozen at −40 °C, and cut to a thickness of 40 µm. All the sections were stained using anti-GFAP (1:1000, Dako, USA) and Nissl antibodies in order to visualize the borders of the contusion injury[10,11,14].

**Analysis of spinal cord damage**. The extent and location of spinal cord damage was evaluated in each rat. The damaged region of the spinal cord was cut in 40 µm thick coronal sections that were stained with an antibody against glial fibrillary acidic protein (GFAP). Spinal cord sections were incubated overnight in serum containing anti-GFAP (1:1000, Dako, USA) antibodies. Immunoreactions were visualized with secondary antibodies labeled with Alexa fluor® 555. Sections were

mounted onto microscope slides using anti-fade fluorescent mounting medium and covered with a cover-glass. The sections corresponding to the lesion epicenter and to the first intact sections immediately rostral and caudal to the injury were selected for each rat, and then imaged using the Olympus Slide Scanner VS120-L100 microscope at 10× magnification. Custom-written Matlab scripts were used to analyze the image. Briefly, the images were divided into square regions of interest (ROI). Files were color-filtered and binarized by means of intensity thresholds that was set empirically and then maintained constant across all the sections. For each lesion core, the number of pixels contained within the regions with spared tissue was calculated. For each spinal cord, this value was compared to the average pixel size of the first intact sections located rostrally and caudally to the lesion. The ratio between both values was used as the amount of spared tissue.

**Statistics**. All other data are displayed as mean values ± SEM. Paired statistical evaluations were performed using Student's $t$-test or the non-parametric Wilcoxon signed-rank test when at least one of the populations could not be assumed to be normally distributed. The Kruskall–Wallis test was applied to all non-paired populations of samples. Tests are one-sided, as our hypotheses were always strictly defined towards the direction of motor improvement. Sample sizes were all equal or superior to $n = 5$ rats. While this sample is often considered small, the use of paired tests and combinatorial properties ensured that the effects of the stimulation could be detected. For non-paired comparisons, we used groups of 6 or 7 rats. All paired analyses were conducted on data that were not collected randomly, as imposed by experimental conditions. Assignment of rats to experimental groups was performed before rehabilitation with the aim to obtain groups with similar initial performance, as quantified in Fig. 10b, c, while ensuring adequate quality of cortical recordings for rats participating in the BSI-enabled training. For kinematic analysis, blinding was not possible. However, tracking of data is a highly automatized process.

**Data availability**. The data that support the findings of this study will be made available by the authors on reasonable request, see author contributions for specific data sets.

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

## Acknowledgements

The authors thank Emanuele Formento for his help developing the experimental protocols and the fruitful discussion on this work's perspectives, Francesca De Cecco, Antoine Philippides, Michèle MacLean, Selin Anil, Laetitia Baud for their essential participation in data acquisition and analysis. The illustration in Fig. 1 was created by Jemère Ruby. This work was supported by a Consolidator Grant from the European Research Council (ERC-2015-CoG HOW2WALKAGAIN 682999), the Wyss center in Geneva, and the Swiss National Science Foundation including the National Centre of Competence in Research in Robotics and the Sinergia program (CRSII3_160696).

## Author contributions

S.M. and G.C. contributed equally to this work; M.B. developed the spinal cord stimulation protocols and brain decoders based on preliminary work from J.D.; M.B. and G.P. performed all the behavioral experiments and analyzed the data; M.B., P.S., and N.P. performed the surgeries; M.B., J.D., S.M., and G.C. conceived the study. G.C. wrote the paper and all the authors contributed to its editing.

## Additional information

**Competing interests:** S.M. and G.C. are founders and shareholders of GTX Medical, a company developing therapies in partial relationships with the topic of the submitted manuscript. M.B., S.M., and G.C. hold a patent on spatiotemporal neuromodulation algorithms (WO2015/063127). The remaining authors declare no competing interests.

