## [Peer Review File · Nature Communications]

Reviewers' comments:

Reviewer #1 (Remarks to the Author):

This study developed and tested a cortical-spine interface in spinal cord-injured rats where recorded signals from motor cortex were used to electrically stimulate epidurally the spinal cord distal to the injury to evoke flexion movements of the hind legs during the swing phase of locomotion. The authors developed two versions of their interface, a binary version that decoded the occurrence or lack thereof of a flexion movement and a proportional version where the magnitude of electrical stimulation was proportional to the cumulative firing rate of a population of M1 neurons. Their main hypothesis is to test whether rehabilitation with their proportional brain-spine interface enhanced recovery from spinal cord injury as compared to continuous spinal stimulation which they have previously shown to be effective. By using two groups of rats who were trained for 5 weeks using either of these stimulation paradigms, they found that proportional brain-spine interface group showed superior performance in various locomotor behaviors.

Major criticisms

1. While the results of this study are impressive, I don't think they have provided "direct proof" that their neurorehabilitation approach has augmented NEUROPLASTICITY and RECOVERY following spinal cord injury as they claim in the abstract. I was getting excited about this prospect when I started reading this but was somewhat disappointed because, as I see it, functional RECOVERY would be demonstrated if it was seen after the brain-spine interface was no longer used and turned off like what was shown by Jackson, Mavoori, and Fetz (2006). In other words, there should be some "semi" permanent plastic changes in the system that sustains recovery even after the interface is removed. This was clearly not demonstrated. Perhaps, however, this is a matter of semantics. Function was shown convincingly to improve over extended use of the interface. So, perhaps, the authors should at least mention the fact that the system would have to be used indefinitely to maintain this recovery, but that might present problems because chronic electrodes fail to record neural signals in the long term.

With regard to NEUROPLASTICITY, again it may be a matter of semantics. Things certainly change in terms of function. But it is not clear that the changes are due to synaptic plasticity and it is also unclear where these changes occur. For example, plasticity could occur within the motor cortex. Or inputs from other cortical areas to motor cortex may have changed as the animal learns to use the interface. They speculate in the Discussion section that use-dependent neuroplasticity occurred among residual connections between the spinal cord and cortex but they don't provide any direct evidence for this, and, therefore, do not provide direct proof that their approach has augmented NEUROPLASTICITY in the synaptic plasticity sense.

2. The text does not refer to figures in an orderly fashion. For example, the authors refer to figure 2A, then 3A and B, then 2C, then figure 4 and 5, and then 3A and B. They should reorder the figure panels so that they are discussed in order.

Minor criticisms

1. I was confused by the terminology of "false-negative rejection rate" and "false-positive rate". Don't the authors mean "correct rejection rate".
2. On several occasions, the authors say "On the average" which should read "On average".
3. In supplementary figure 3a, the foot-off decoder triggers stimulation at a point that is different from the threshold crossing. What is the difference between the "foot-off decoder trigger" and the threshold crossing?

Reviewer #2 (Remarks to the Author):

The paper describes the use of epidural spinal stimulation triggered by ensemble activity in motor cortex to enhance leg flexion during induced bipedal locomotion in rats with a severe thoracic spinal contusion injury. The results include the first demonstration of a proportional brain-spinal interface in which the amplitude of a decoded cortical signal controls the amplitude of continuous spinal stimulation to produce graded flexion movements. Although this finding is novel, it is a relatively incremental advance over previous work in spinal cord injured rats, monkeys, and humans from the Courtine and other labs, all cited in the paper. Therefore, it will be primarily of interest to specialists in the brain-computer interface field.

The study design is sound, the methods are state-of-the-art, and the experiments are well-executed. The statistical analyses are appropriate and valid. The paper leaves out many experimental details which would make it difficult for others to reproduce the experiments (see Minor Comments).

Major Comments

1. pg. 5, section on the proportional brain-spine interface. The text and figures do not discuss or show, respectively, EMG during operation of the proportional brain-spine interface. The relationship between the variable stimulation amplitude and EMG, especially in comparison to the EMG during continuous stimulation and the binary interface, is important to understand how the interface improves stepping and should be presented in a figure and discussed.
2. The main new finding of the paper is that a proportional brain-spine interface improves locomotor performance more than a binary interface, since the Courtine lab previously demonstrated (Capogrosso et al 2016) in monkeys with a spinal cord injury that a similar binary interface improves locomotion. However, the key comparisons in Figures 7 and 8 are between the proportional interface and continuous stimulation. This reduces the impact of these results in relation to the Capogrosso et al paper.
3. The final results section shows that 5 weeks of training with the proportional interface produces improvements in functional recovery that do not require ongoing use of the proportional stimulation. The Discussion uses this result to argue that the repeated training with the interface produces neural plasticity that has clinical significance. Although the training undoubtedly leads to some plasticity, it is unclear how important this result is for translational research. Even after 5 weeks of training with the proportional interface, the rat presumably cannot step without administration of serotonin agonists and delivery of at least

continuous stimulation. Therefore, it does not appear that there was enough plasticity to support significant behavioral recovery without continued use of one of the stimulation interfaces.

4. Another factor that tempers the significance of the results regarding development of a clinical neuroprosthetic is the nature of the modulated cortical activity. As suggested by the authors, the activity in the rat is likely driven by sensory feedback. Such a signal is probably not useful for controlling a brain-spinal interface for restoring locomotion in humans where a volitional cortical signal is needed to generate a wider repertoire of movements.

Minor comments

1. pg. 3, line 28. Define the parenthetical quantification of the swing phase, i.e., "18.2 ± 5.3 %" of what?

2. pg. 4, line 11. The use of the term "inferior" is not clear. Is "smaller" what is intended?

3. pg. 4, 3rd paragraph. It is not clear how accurate the thresholded normalized cumulative firing predicted the "100 ms anticipation" of foot-off times. Data are presented on the success of the decoder in predicting foot-off times, but not how well it predicts the time 100 ms before foot-off. In the next paragraph, the average time of stimulation onset relative to foot-off is reported, but it would be interesting to know the distribution of predicted 100 ms anticipation times without stimulation. This will indicate how well a single threshold predicted the 100 ms anticipation events from step to step.

4. Fig. 3 legend. The legend for Step 4A states that stimuli were delivered when the normalized cumulative firing crossed a threshold corresponding to 200 ms before foot-off – the text in the Results and Methods sections states 100 ms.

5. Figs. 3c, 7c,d. The shaded regions around the average traces are presumably a measure of variance, but this should be specified in the legends.

6. pg. 4, paragraph 4. Indicate the stimulus current that was used for the binary brain-spine interface.

7. Fig. 4a shows an example similar to Fig. 2a and is probably not necessary.

8. Fig. 5a legend. How was it determined that the evoked potentials in the shaded area were due to direct stimulation of the motor nerve?

9. Fig. 5b legend. Specify the measure of response amplitudes that was used.

10. pg. 4, last paragraph. Indicate whether these correlations were calculated with no stimulation, continuous spinal stimulation, or the binary-spine interface.

11. pg. 5, 2nd to last paragraph. The term "step height" should be used rather than "locomotor performance" since it is the only parameter shown in Fig. 6B.

12. pg. 5, last paragraph. Indicate the time after injury that the stair climbing task was performed.
13. Fig. 7b legend. Define a "tumble."
14. pg.6, section on rehabilitation enabled by the proportional brain-spine interface. Indicate whether the animals were trained on the treadmill or overground.
15. pg. 12, line 15. The Precision Systems and Instrumentation device is called the "Infinite Horizon Impactor."
16. pg. 12, line 5 from the bottom. Please add another citation to the paper(s) that best describe(s) the correlation method.
17. pg. 13, section on cortical recordings. The description of the multi-unit recordings seems unnecessarily complicated. I guess that a spike-time event was recorded whenever the signal rose through the threshold level. If so, it seems confusing to use the phrase "field potential stochastic events."
18. pg. 13, section on cortical decoding. Please describe the method used to determine which 6 MUA channels "correlated most" with muscle activity.
19. pg. 13, section on cortical decoding. More information on the calculation of the normalized cumulative firing would be helpful. In the figures, the normalized cumulative firing appears to be a continuous variable, or at least has more amplitude and temporal resolution than could be obtained by binning 6 channels of spike events. Please clarify how the channels were normalized, how they were combined, and how the weights were determined for the linear combination.
20. pg. 13, section on binary-spine interface. The term "flexion detections" is not defined (I guess it's equivalent to "event detection" but this should be clear).
21. pg. 14. section on analysis of spinal cord damage. Were the sections labeled with a cell body or myelin stain? The description of the measurement ("The amount of spared tissue was computed as the ratio between the number of pixels at the epicenter and in the intact sections.") is unclear; please provide additional information.
22. Supplementary Fig. 3c,d. Can the mean values for the individual rats be plotted for the gait rehabilitation as for continuous stimulation and binary brain-spine interface?

NCOMMS-17-29883

BRAIN-CONTROLLED MODULATION OF SPINAL CIRCUITS IMPROVES RECOVERY FROM SPINAL CORD INJURY

We would like to thank all the reviewers for their appreciation of the advances brought by our study, and for the useful comments. In particular, we realized that we had not sufficiently emphasized the central objective of our study: develop a brain-controlled modulation therapy that combines all the technical (effective facilitation of locomotion compared to other controllers, task-specific modulation of leg movements) and practical (no training prior to injury, rapid calibration to enable daily rehabilitation) features that are necessary to deploy such complex technology every day in rats with severe spinal cord injury. This refocus required the addition of new experiments and analyses, which resulted in several new figures and panels in the revised version of the manuscript.

We also removed references to neuroplasticity. Indeed, although the improvements brought by the brain-spine interface-based training necessarily involved plastic changes, the present study did not uncover these changes. Such evaluations would require extensive anatomical studies that are beyond the scope of this study.

In addition, we have carefully taken all your suggestions into consideration to prepare a revised version of the manuscript, which required additional data processing and the preparation of new figures.

To facilitate your task in navigating through all these changes, we have implemented the following colour coding system to respond to your queries, and explain the related changes in the manuscript:

AUTHORS' ANSWERS: Answer to specific questions.

ACTION IN THE MANUSCRIPT: Actions in the manuscript (in the revised text as **bold red characters**).

We feel that your comments and suggestions have contributed to improving the quality and clarity of the manuscript, and we therefore would like to thank you for the time you spent in reviewing our work.

Reviewer #1

GENERAL COMMENTS:

REVIEWER's COMMENT: This study developed and tested a cortical-spine interface in spinal cord-injured rats where recorded signals from motor cortex were used to electrically stimulate epidurally the spinal cord distal to the injury to evoke flexion movements of the hind legs during the swing phase of locomotion. The authors developed two versions of their interface, a binary version that decoded the occurrence or lack thereof of a flexion movement and a proportional version where the magnitude of electrical stimulation was proportional to the cumulative firing rate of a population of M1 neurons. Their main hypothesis is to test whether rehabilitation with their proportional brain-spine interface enhanced recovery from spinal cord injury as compared to continuous spinal stimulation which they have previously shown to be effective. By using two groups of rats who were trained for 5 weeks using either of these stimulation paradigms, they found that proportional brain-spine interface group showed superior performance in various locomotor behaviors.

While the results of this study are impressive, I don't think they have provided "direct proof" that their neurorehabilitation approach has augmented NEUROPLASTICITY and RECOVERY following spinal cord injury as they claim in the abstract. I was getting excited about this prospect when I started reading this but was somewhat disappointed because, as I see it, functional RECOVERY would be demonstrated if it was seen after the brain-spine interface was no longer used and turned off like what was shown by Jackson, Mavoori, and Fetz (2006). In other words, there should be some "semi" permanent plastic changes in the system that sustains recovery even after the interface is removed. This was clearly not demonstrated. Perhaps, however, this is a matter of semantics. Function was shown convincingly to improve over extended use of the interface. So, perhaps, the authors should at least mention the fact that the system would have to be used indefinitely to maintain this recovery, but that might present problems because chronic electrodes fail to record neural signals in the long term.

AUTHORS' ANSWERS: As Reviewer 2 pointed out, "the training undoubtedly leads to some plasticity". For this reason, we combined the concept of neuroplasticity and recovery. However, we agree with you that these two concepts are not linked with quantified data and causal experiments in the current manuscript. The paper is exclusively focused on the development of a brain-spine interface that combines the technical and practical features necessary to be deployed during training, and the evaluation of the impact of training with this neuroprosthesis on functional recovery compared to continuous stimulation. In this study, we did not assess the anatomical reorganization of the descending pathways after training, for multiple reasons:

1- We just published a manuscript at Nature Neuroscience in which we studied the anatomical and functional reorganization of residual connections and spared circuits in response to gait rehabilitation with continuous stimulation (Asboth et al., Nature Neuroscience 2018). This work required extensive and complex experiments that would be very difficult to implement in combination with the brain-spine interface. Contrary to the studies conducted by Dr. Fetz in which the plasticity is restricted to the neurons / circuits that are directly linked with the electronic neural implant, the brain-spine interface likely triggers plastic changes in a variety of locations. Indeed, compared to studies by Dr. Fetz, the brain-spine interface not only linked broader regions of the central nervous system (M1 and lumbar spinal cord) but also enabled the execution of complex motor functions involving multiple circuits and neural pathways, i.e. way beyond the two inter-connected regions with the brain-spine interface. Consequently, the neural changes underlying the improvements that are specifically mediated by the brain-spine interface during gait rehabilitation (beyond training versus no training that we documented extensively) are difficult to identify.

2- We feel that it is not surprising that after a severe spinal cord contusion that leads to complete and permanent paralysis, the recovery of meaningful function would still require the neuroprosthesis. As you pointed out, however, gait rehabilitation enabled the rats to learn how to use the neuroprosthesis for executing basic and complex locomotor tasks, suggesting the occurrence of substantial plastic changes to support these improvements and skilled use of the prosthesis. Clearly, the reorganization of spared circuits and residual pathways mediated by 5 weeks of training was not sufficient to restore

motor control without the neuroprosthesis. However, this specific issue depends on the severity of the injury, duration of training, and task-specific training conditions. With a longer and more extensive training in both bipedal and quadrupedal locomotion (9 weeks, twice per day), we found that injured rats could regain meaningful locomotor capacities without any assistance and stimulation (Asboth et al., Nature Neuroscience 2018). Here, we did not aim at obtaining the more extensive recovery possible. For example, we limited the training to 5 weeks in order to ensure the maintenance of neural signals with sufficient quality throughout training (see below for your comment on neural signal longevity). Our aim was to evaluate whether, when training rats in the same conditions, a direct cortical control of spinal cord stimulation during gait rehabilitation leads to superior recovery compared to continuous stimulation. To our knowledge, no previous studies had been able to conduct rehabilitation with an implanted neural bypass after neurological disorders. Our brain-spine interface was designed to enable such studies, i.e. we conceived a brain-spine interface that did not require training before the injury to be operated and required minimal calibration for everyday use during rehabilitation.

3- We are conducting a clinical study in people with chronic spinal cord injury who are trained with electrical spinal cord stimulation (no brain control) and the gravity-assist. Overtime, all the enrolled participants exhibited robust improvements of their locomotor capacities, both with and without the stimulation. These results, both in rats and humans, indicate that gait rehabilitation enabled by electrical spinal cord stimulation mediates plastic changes that improve motor control both without and with stimulation. The improvements with stimulation are important, since the participants of our clinical study are able to use the neuroprosthesis to ambulate outside the laboratory environment. Yet, all the participants of the clinical study who completed the rehabilitation program expressed the desire to have more control over the stimulation protocols in their daily life. This volitional control over stimulation protocols would only be possible with a brain-spine interface. While we cannot discuss these additional results in humans in this manuscript, we feel that we should be careful in the discussion of the presented results, since we already have gained an extensive knowledge on the potential of gait rehabilitation enabled by neuromodulation therapies to improve motor functions after spinal cord injury.

Since we very much agree with the general scope of your comment, we took several actions in order to address this issue and better highlight our goals and the results supporting them.

ACTION IN THE MANUSCRIPT: First, we removed the term neuroplasticity from the Abstract, focusing the message on the functional outcome of the brain-spine interface and gait rehabilitation. In this process, we reformulated the main hypothesis of the study:

Introduction: Here, we sought to test the hypothesis that, compared to continuous spinal cord stimulation, a direct cortical control over adaptive spinal cord stimulation protocols during gait rehabilitation enhances functional recovery from a severe, clinically-relevant spinal cord injury.

Second, we rewrote the introduction to better highlight the complex issues of deploying a neural bypass every day during gait rehabilitation, and that our technology effectively addressed these issues:

Abstract: The delivery of brain-controlled neuromodulation therapies during motor rehabilitation may augment recovery from neurological disorders. To test this hypothesis, we conceived a brain-controlled neuromodulation therapy that combines all the technical and practical features necessary to be deployed daily during gait rehabilitation. Rats received a severe spinal cord contusion that led to leg paralysis. We engineered a brain-spine interface whereby cortical ensemble activity constantly determined the intensity of spinal cord stimulation protocols promoting leg flexion during swing. After minimal calibration time and without prior training, this neural bypass enabled paralyzed rats to walk overground and adjust foot clearance in order to climb a staircase. Compared to continuous spinal cord stimulation, brain-controlled stimulation accelerated and enhanced the long-term recovery of locomotion. These results demonstrate the relevance of brain-controlled neuromodulation therapies to augment recovery from motor disorders, establishing important proofs-of-concept that warrant clinical studies.

Third, we added a section on the added value of the brain-spine interface to support complex locomotor tasks after gait rehabilitation and addressed your concerns regarding the problem of chronic neural recordings:

The proportional brain–spine interface enabled the rats to regain volitional control over task-specific leg movements but even after gait rehabilitation, the rats were dependent on the presence of stimulation to produce robust locomotor movements. After less severe injuries and/or more extensive training regimen, recovery can occur without any stimulation. However, the availability of a practical brain–spine interface would still be desirable for the more severely affected patients, indicating the importance of developing long-lasting prosthetic technologies for clinical applications. To this respect, recordings of multiunit activity from intracortical neural probes extinguish rapidly, which would prevent the long-term use of a proportional brain–spine interface operating with such probes in humans. However, the robustness of the proportional controller came from the relative simplicity of the control signal—a simple summation of multiunit activity recorded from cortical ensemble population that required minimal daily recalibration. Accordingly, local field potentials, which allowed human participants to control brain–machine interfaces for several years³², may be sufficient for operating a proportional brain–spine interface.

Fourth, to emphasize the novelty of the developed brain-spine interface, we added a section on the stringent requirements for the development and implementation of a brain-spine interface that could support long-lasting gait rehabilitation:

Introduction: Addressing this hypothesis involves a series of engineering challenges. First, clinically-relevant settings require the conception of a brain-spine interface that does not require training prior to injury. Second, the time required for calibrating the brain-spine interface must be minimal in order to support daily sessions of gait rehabilitation. Third, due to importance of task-specific rehabilitation, the brain-spine interface must enable training in natural locomotor tasks such as overground walking and stair climbing, not only during automated stepping on a treadmill¹⁴. To tackle these challenges, we took advantage of our previous developments, both the advanced spinal cord stimulation protocols elaborated in rodents¹⁵⁻¹⁷ and the brain-computer interface technologies conceived in nonhuman primates³.

Fifth, we stressed the easy use and robustness of the brain-spine interface for daily gait rehabilitation in the Result section:

Results: Over the entire duration of the rehabilitation program, the daily calibration of the brain–spine interface only required a few minutes of fine-tuning and could then be operated with minimal changes throughout the 30 min training sessions.

Sixth, we reemphasized the clinical importance of this result in the Discussion:

Discussion: This ecological approach²⁸ enabled a rapid calibration of the neural bypass, and its immediate use for the production of adaptive locomotor movements and daily gait rehabilitation. **These features are essential for the practical deployment of a brain–spine interface that can support gait rehabilitation every day in clinical settings.**

REVIEWER's COMMENT: With regard to NEUROPLASTICITY, again it may be a matter of semantics. Things certainly change in terms of function. But it is not clear that the changes are due to synaptic plasticity and it is also unclear where these changes occur. For example, plasticity could occur within the motor cortex. Or inputs from other cortical areas to motor cortex may have changed as the animal learns to use the interface. They speculate in the Discussion section that use-dependent neuroplasticity occurred among residual connections between the spinal cord and cortex but they

don't provide any direct evidence for this, and, therefore, do not provide direction proof that their approach has augmented NEUROPLASTICITY in the synaptic plasticity sense.

AUTHORS' ANSWERS: As you pointed out, the brain-spine interface directly linked the motor cortex to the spinal cord. However, the execution of locomotion engaged a multiplicity of circuits and neural pathways, beyond the motor cortex and spinal cord. This strategy enabled a cooperation between the natural neural control of leg movements (residual connections) and brain-spine interface-mediated modulation of the spinal cord. Consequently, we are fully aligned with your comments that the identification of the location of the plastic changes underlying the improvements over time is unlikely to be straightforward, since they likely take place at multiple locations. Indeed, our recent study showed that gait rehabilitation mediates an extensive reorganization of cortical, brainstem and spinal circuits, suggesting that the additional effects of the brain-spine interface may occur at any of these locations, but also / or in additional locations. We are currently combining calcium imaging in freely behaving rats with virus-mediated tract tracing, optogenetics, and chemogenetics to tackle this important scientific question. This study started two years ago but will still require several years of work to be completed. Preliminary results indeed suggest that the thalamus and cerebellum might play a key role in mediating the additional improvement with the brain-spine interface. However, we feel that the study of the neural mechanisms underlying the therapeutic impact of the proportional brain-spine interface goes beyond the scope of the present study. While there is no doubt that the brain-spine interface-based gait rehabilitation triggers plasticity, we fully agree that our manuscript does not expose this plasticity. Consequently, we adjusted the message to avoid stating that our procedure mediated plasticity.

ACTION IN THE MANUSCRIPT: We have restricted references to potential plastic changes to the Introduction and Discussion sections. In the discussion, we toned down the reference to plastic changes.

For example, we introduce the possibility of plastic changes in a more conservative fashion:

Discussion: We propose that this property may have triggered use-dependent neuroplasticity of residual connections during rehabilitation.

We also change / adapted the section of the Discussion on plastic changes:

We propose that, both in rats and humans, these gait rehabilitation programs closing the loop between circuits located above and below the injury increase use-dependent neuroplasticity of residual connections^{8,10,14}. Bidirectional spike-timing-dependent neuroplasticity would be the most probable mechanism steering this reorganization⁶⁻⁸. However, this interpretation remains speculative. The physiological, anatomical and molecular mechanisms that may support or invalidate this potential explanation will require additional studies.

REVIEWER's COMMENT: The text does not refer to figures in an orderly fashion. For example, the authors refer to figure 2A, then 3A and B, then 2C, then figure 4 and 5, and then 3A and B. They should reorder the figure panels so that they are discussed in order.

AUTHORS' ANSWERS: We indeed explained the technical design of both brain-spine interface within the same figures, which led to the need to refer frequently to Figure 3.

ACTION IN THE MANUSCRIPT: We split Figure 3 into two figures. Moreover, we proposed additional figures to respond to Reviewer 2. In this process, we have carefully organized the figures and panels in order to refer to the figures in an orderly fashion.

REVIEWER's COMMENT: I was confused by the terminology of "false-negative rejection rate" and "false-positive rate". Don't the authors mean "correct rejection rate".

AUTHORS' ANSWERS: We agree with the reviewer that the terminology previously used could lead to confusion for the reader.

ACTION IN THE MANUSCRIPT: A clearer terminology consisting of {foot-off detected, missed detection, false detection, true negative} have been applied throughout the manuscript and in all the associated figures.

REVIEWER's COMMENT: On several occasions, the authors say "On the average" which should read "On average".

AUTHORS' ANSWERS: Many thanks for pointing out this mistake

ACTION IN THE MANUSCRIPT: We corrected this mistake.

REVIEWER's COMMENT: In supplementary figure 3a, the foot-off decoder triggers stimulation at a point that is different from the threshold crossing. What is the difference between the "foot-off decoder trigger" and the threshold crossing

AUTHORS' ANSWERS: The binary decoder combines two crossing rules that consist in threshold crossing and a certain refractory period between detections, as described in the section Binary brain–spine interface of the Methods. What the reviewer signaled is an instance of the operation of this second decoding rule, which slightly delayed the detection of the foot-off event.

ACTION IN THE MANUSCRIPT: Figure 3 describes the calibration and operations of the binary brain-spine interface. We have added labels to point out the importance of the refractory period in order to prevent false detections.

Reviewer #2

GENERAL COMMENTS:

The paper describes the use of epidural spinal stimulation triggered by ensemble activity in motor cortex to enhance leg flexion during induced bipedal locomotion in rats with a severe thoracic spinal contusion injury. The results include the first demonstration of a proportional brain-spinal interface in which the amplitude of a decoded cortical signal controls the amplitude of continuous spinal stimulation to produce graded flexion movements. Although this finding is novel, it is a relatively incremental advance over previous work in spinal cord injured rats, monkeys, and humans from the Courtine and other labs, all cited in the paper. Therefore, it will be primarily of interest to specialists in the brain-computer interface field. The study design is sound, the methods are state-of-the-art, and the experiments are well-executed. The statistical analyses are appropriate and valid. The paper leaves out many experimental details which would make it difficult for others to reproduce the experiments (see Minor Comments).

AUTHORS' ANSWERS: Many thanks for your appreciation of the efforts deployed to implement this study. We agree that the proportional controller will primarily be of interest for neuroprosthetic community. However, this community has grown tremendously over the past decade. We feel that our gait neuroprosthesis will meet the broad interest of this community, since no previous studies could maintain a direct and continuous link between cortical activity and spinal cord stimulation protocols in order to enable a fluid control over a broad range of leg movements in otherwise paralyzed animal models. The robustness of this controller was such that we could train otherwise paralyzed animals every day with this brain-spine interface, without extensive recalibration or complex learning procedures prior to or after the injury. It is also important to point out that in our previous work in nonhuman primates, the brain-spine interface was binary. When turning on the brain-spine interface, the monkeys regained basic locomotor movements, but they were not able to adjust the trajectory of their leg in order to meet task-specific requirements. In the revised version of the manuscript, we show that the ankle muscle activity could potentially be used as a surrogate for brain signals in order to trigger such stimulation protocols. However, we also show that the rats are able to utilize the brain-spine interface to mediate volitional change in the trajectory of their otherwise paralyzed legs. In the revised version of the manuscript, we show that this volitional modulation is contingent on brain signals. Therefore, the proportional controller represents more than an incremental advance, since the direct control over flexor muscle activity opens novel perspectives on the potential of brain-spine interfaces to restore a graded control over leg movements after neurological deficits.

In addition to the neuroprosthetic community, the superior effect of rehabilitation with the proportional brain-spinal interface compared to the current state of the art (continuous stimulation) will be of interest for a much broader community, spanning rehabilitation specialists, spinal cord injury medicine, and even scientists focusing on learning and plasticity.

Consequently, we feel that this study is more than an incremental improvement of existing brain-machine interface and rehabilitation paradigms. Many reviews have been written recently on the intriguing possibility to increase recovery with brain-machine interface driven gait rehabilitation. Our study provides evidence supporting this hypothesis.

ACTION IN THE MANUSCRIPT:

We have modified the Abstract in order to highlight the complexity of the technology that had to be developed to address our main question:

The delivery of brain-controlled neuromodulation therapies during motor rehabilitation may augment recovery from neurological disorders. To test this hypothesis, we conceived a brain-controlled neuromodulation therapy that combines all the technical and practical features necessary to be deployed daily during gait rehabilitation. Rats received a severe spinal cord contusion that led to leg paralysis. We engineered a brain-spine interface whereby cortical ensemble activity constantly determined the intensity of spinal cord stimulation protocols promoting leg flexion during swing. **After minimal calibration time and without prior training,** this neural bypass enabled paralyzed rats to walk overground and adjust foot clearance in order to climb a staircase. Compared to continuous spinal cord stimulation, brain-controlled stimulation accelerated and enhanced the long-term recovery of locomotion. These results demonstrate the relevance of brain-controlled neuromodulation therapies to augment recovery from motor disorders, establishing important proofs-of-concept that warrant clinical studies.

Additional changes are described below.

REVIEWER's COMMENTS: pg. 5, section on the proportional brain-spine interface. The text and figures do not discuss or show, respectively, EMG during operation of the proportional brain-spine interface. The relationship between the variable stimulation amplitude and EMG, especially in comparison to the EMG during continuous stimulation and the binary interface, is important to understand how the interface improves stepping and should be presented in a figure and discussed.

AUTHORS' ANSWERS: Many thanks for pointing out the importance of these analyses. We initially took these results out to limit the length of the manuscript, but we concur on their relevance to understand how the proportional controller modulates leg trajectory. We also added a series of analyses to point out the critical importance of the proportional controller to support task-specific training.

ACTION IN THE MANUSCRIPT:

1- We have significantly reorganized the figures to better document the features and performances of the proportional brain-spine interface. For this purpose, we expanded Figure 6, which now better explains the configuration of the proportional brain-spine interface (panel A) and reports an extensive analysis of the modulation of the EMG activity. Specifically, in panel C, we propose a succession of zooms on the bursts of EMG activity in order to explain how the changes in the intensity of the EES pulses determines the modulation of the reflex responses, and thereby the overall EMG activity. We also report the correlations between the cortical activity and the EMG bursts, and show that cortical activity explained a high percentage of the variance in EMG activity during locomotion with the proportional brain-spine interface compared to continuous stimulation.

The description of these results in the Results section reads as follows:

A new group of 5 rats participated to these experiments. As early as 7 days after injury, all the rats showed the expected modulation of cortical ensemble population during locomotion (Fig. 6B). Without prior training, the proportional brain–spine interface enabled these rats to modulate the amplitude of ankle flexor muscle activity using neural signals directly recorded from the motor cortex (Fig. 6C). Examination of EMG bursts in the ankle flexor muscle revealed that muscle activity was elaborated from a succession of motor responses that were linked to each pulse of stimulation. Since the amplitude of each pulse of stimulation was proportional to the instantaneous cumulating firing of cortical ensemble population, the amplitude of muscle activity was linearly correlated with motor cortex activity (Fig. 6D-E).

We therefore evaluated whether the proportional brain–spine interface restored the relationship between motor cortex activity and step height that otherwise only emerges after several weeks of gait rehabilitation. As expected, early after injury there was no correlation between the cortical ensemble population and the step height when rats stepped automatically with continuous stimulation ($R^2 = 0$ % of explained variance; Fig. 6F). The forced link between cortical ensemble population and flexor motoneuron activity re-established this relationship (Fig. 6F). At 10 days post-injury, cortical ensemble activity at foot-off determined 42.2 ± 4.7 % of the variance in step height during the subsequent step ($P = 0.03$, Fig. 6G), which was comparable to the values measured in rats after 5 weeks of training enabled by continuous stimulation and the gravity-assist ($R^2 = 49.9 \pm 6.9$ %; Fig. 6G).

These results show that shortly after injury and without any training, the proportional brain–spine interface established a contingency between ankle flexor muscle activity and cortical ensemble firing that determined the height of each step during locomotion of otherwise paralyzed rats.

2- We added a dedicated section in the Results “ **The proportional brain–spine interface improves locomotor performance** “ in which we report additional analyses on locomotor performance with continuous stimulation, binary control of stimulation and proportional control of stimulation. These results are reported in the new Figure 7. The Results reads as follows:

We then asked whether the proportional brain–spine interface improved locomotor performance compared to both continuous stimulation and binary control of stimulation.

Locomotor performance was evaluated during overground locomotion with the gravity–assist (Fig. 7A). To quantify locomotor performance, we applied a principal component

analysis¹⁵ to a large number of parameters calculated from kinematic recordings ($n = 55$ parameters). Locomotor performance was quantified as the distance from intact rats in the space defined by the first three principal components (explained variance, 63.0 %).

Compared to both continuous stimulation and binary control of stimulation, the proportional brain–spine interface enabled rats to produce gait patterns that resembled more closely those recorded in intact rats ($P < 0.05$, Fig. 7B). In particular, the proportional control over stimulation protocols significantly reduced the amount of paw dragging ($P < 0.001$, Fig. 7C), increased the distance covered overground per unit of time ($P = 0.06$, Fig. 7D) and improved the coordination between the oscillations of lower limb segments (Fig. 7E). Instead, the sudden burst of stimulation triggered by the binary controller tended to disrupt intra-limb coordination (Fig. 7E).

These combined analyses demonstrate the physiological and functional superiority of the proportional brain–spine interface to enable locomotion after a severe spinal cord injury.

3- We improved the analyses and representation of the stair climbing task, in order to better emphasize the need of the proportional controller to enable the volitional modulation of leg movements to pass the staircase. In this process, we created a new section dedicated to this task in which we better introduced the goal of this experiment:

We next evaluated whether the rats were able to exploit the proportional brain–spine interface to achieve task-specific modulation of leg movements. For this purpose, we tested the rats one month after injury during a stair climbing task that required a voluntary increase in foot clearance during swing (Fig. 8A).

4- We dedicated a new section and a Figure (Figure 9) on the critical importance of the proportional controller to enable task-specific modulation of leg movements. Specifically, we show that the binary controller cannot be used to support a volitional adjustment of leg movements, since the rats exert no control over the parameters of stimulation. We illustrate that other physiological features such as the EMG activity of leg muscles are inappropriate to enable a volitional control of task-specific leg movements. The new Result section reads as follows:

Volitional modulation of gait is contingent on brain–controlled stimulation

We then studied the contingency on the activity of cortical neurons for the delivery of proportional stimulation protocols that support the volitional modulation of leg movements. Specifically, we asked whether more accessible physiological signals could be harnessed to detect foot–off events and modulate stimulation protocols to achieve a desired step height.

For this purpose, we selected the EMG activity of the ankle flexor muscle. As early as 10 days after injury, we found that ankle muscle activity enabled decoding foot–off events with a high degree of accuracy, which was comparable to the levels obtained with cortical activity (Fig. 9A). This result suggest that a controller based on ankle muscle activity may be sufficient to operate a binary controller.

We then studied the possibility to decode the step height from ankle muscle activity. While the increase in the cumulative firing of cortical ensemble population anticipated the vertical displacement of the foot, the EMG burst was logically concomitant to foot movements (Fig. 9B). Consequently, the amplitude of the subsequent step height could not be decoded from the activity of the ankle muscle at foot–off (Fig. 9C). Moreover, during locomotion enabled by continuous stimulation, ankle muscle activity failed to predict the cumulative firing of cortical ensemble population, both early and late after the injury (Fig. 9D).

We concluded that ankle muscle activity could not be used as a surrogate of cortical activity to allow the rats to tune stimulation protocols in order to modulate leg movements voluntarily, thus highlighting the importance of the proportional brain–spine interface for task-specific gait adjustments.

REVIEWER's COMMENTS: The main new finding of the paper is that a proportional brain-spine interface improves locomotor performance more than a binary interface, since the Courtine lab previously demonstrated (Capogrosso et al 2016) in monkeys with a spinal cord injury that a similar binary interface improves locomotion. However, the key comparisons in Figures 7 and 8 are between the proportional interface and continuous stimulation. This reduces the impact of these results in relation to the Capogrosso et al paper.

AUTHORS' ANSWERS: We realize that we did not explicitly point out the limitations of the previous work in nonhuman primates. We have not been able to deliver the brain-spine interface developed in nonhuman primates during training. We agree that the previous version of the manuscript failed to emphasize the key new features of the work. We took several actions to remedy this issue.

ACTION IN THE MANUSCRIPT: We added an extensive section in the Introduction that clearly points out the results that we were previously obtained in rodent models and nonhuman primates, and their limitations to test our main hypothesis: gait rehabilitation enabled by a brain-controlled modulation therapy.

Introduction: Addressing this hypothesis involves a series of engineering challenges. First, clinically-relevant settings require the conception of a brain-spine interface that does not require training prior to injury. Second, the time required for calibrating the brain-spine interface must be minimal in order to support daily sessions of gait rehabilitation. Third, due to importance of task-specific rehabilitation, the brain-spine interface must enable training in natural locomotor tasks such as overground walking and stair climbing, not only during automated stepping on a treadmill¹⁴. To tackle these challenges, we took advantage of our previous developments, both the advanced spinal cord stimulation protocols elaborated in rodents¹⁵⁻¹⁷ and the brain-computer interface technologies conceived in nonhuman primates³.

We previously showed that the delivery of epidural electrical stimulations over specific spinal cord locations and at a precise timing finely modulates the degree of extension and flexion of the paralyzed legs in rats with severe spinal cord injury¹⁵⁻¹⁷. **However, the rats have no control over the stimulation.** Consequently, the modulation of leg movements remains involuntary—preventing the use of these stimulation protocols to encourage voluntary motor control during gait rehabilitation. **Inversely, we reported that nonhuman primates can immediately operate a brain–spine interface to execute basic locomotor movements of a paralyzed leg, but they are not able to modulate the stimulation protocols in order to adapt leg movements to task-specific requirements³. Due to these limitations, the therapeutic potential of gait rehabilitation enabled by a brain-spine interface to augment recovery from spinal cord injury has not been evaluated.**

Results: The new Figure 7 shows the immediate impact of the three stimulation conditions on locomotor performance, demonstrating that the proportional controller mediates improved gait features compared to the two other paradigms. We also show with the new Figure 8 and Figure 9 that the proportional controller is necessary to enable task-specific adjustment of leg movements, which is necessary to deliver activity-based rehabilitation: we pointed this out in the Introduction:

Introduction: Third, due to importance of task-specific rehabilitation, the brain-spine interface must enable training in natural locomotor tasks such as overground walking and stair climbing, not only during automated stepping on a treadmill¹⁴.

Consequently, we feel that it is more logical and sufficient to compare the impact of gait rehabilitation between the current state-of-the-art training conditions (continuous stimulation) and the more efficient and versatile brain-spine interface technology (Proportional brain-spine interface). From our experience, each batch of rats show slightly different features and recovery. Therefore, we always compare experimental conditions on the same batch of rats and during parallel studies, to ensure the uniformity of experimental conditions for all the experimental groups. Adding a third group of rats trained with the binary brain-spine interface would thus require performing a study with the three

groups. We feel that this extensive amount of work would not add much information in comparison of the invested effort and resource. We added a sentence in the Results to point this out:

We did not add a group of rats trained with the binary brain–spine interface since this paradigm was inferior to the proportional brain–spine interface to improve gait patterns and did not enable task-specific training.

REVIEWER's COMMENTS: The final results section shows that 5 weeks of training with the proportional interface produces improvements in functional recovery that do not require ongoing use of the proportional stimulation. The Discussion uses this result to argue that the repeated training with the interface produces neural plasticity that has clinical significance. Although the training undoubtedly leads to some plasticity, it is unclear how important this result is for translational research. Even after 5 weeks of training with the proportional interface, the rat presumably cannot step without administration of serotonin agonists and delivery of at least continuous stimulation. Therefore, it does not appear that there was enough plasticity to support significant behavioral recovery without continued use of one of the stimulation interfaces.

AUTHORS' ANSWERS: Reviewer 1 made a similar comment. Please read the response to his/her comment (first two comments), and all the associated changes in manuscript. In addition, we specifically added a section in the Discussion.

ACTION IN THE MANUSCRIPT:

Discussion: The proportional brain–spine interface enabled the rats to regain volitional control over task-specific leg movements but even after gait rehabilitation, the rats were dependent on the presence of stimulation to produce robust locomotor movements. After less severe injuries and/or more extensive training regimen, some recovery can occur without any stimulation³². However, the availability of a practical brain–spine interface would still be desirable for the more severely affected patients, indicating the importance of developing long-lasting prosthetic technologies for clinical applications.

REVIEWER's COMMENT: Another factor that tempers the significance of the results regarding development of a clinical neuroprosthetic is the nature of the modulated cortical activity. As suggested by the authors, the activity in the rat is likely driven by sensory feedback. Such a signal is probably not useful for controlling a brain–spinal interface for restoring locomotion in humans where a volitional cortical signal is needed to generate a wider repertoire of movements.

AUTHORS' ANSWERS: We agree with this important point.

ACTION IN THE MANUSCRIPT: We have added a section on this issue in the Discussion:

Theoretically, the continuous proportional controller developed in rats could translate into phase–dependent control algorithms that continuously modulate the degree of extension and flexion for each leg. Such a state–dependent, proportional brain–spine interface has the potential to mediate a markedly more refined prosthetic control of the legs.

The viability of this control strategy is contingent on the similarities between the source of cortical modulation in rodents and primates. As we discuss below, our results suggest that the modulation of cortical activity during stepping on a treadmill was essentially driven by sensory feedback. Instead, nonhuman primates displayed modulation of cortical activity prior to initiating leg movements, suggesting that cortical ensemble population encodes motor intentions. However, we showed that rats also learned to modulate cortical activity in order to mediate intentional increases in foot clearance to climb a staircase. As in nonhuman primates, this modulation occurred prior to the desired change in leg trajectory, suggesting that this task-specific modulation reflected motor intentions. Finally, two decades of research in brain–computer interface showed that people with long-lasting tetraplegia are capable of modulating leg motor cortex activity without overt movement²⁹. They can also control robotic arms^{30,31} and neuromuscular stimulators^{1,2} to execute manual tasks using only signals recorded from the motor cortex. These results suggest that the relatively simple decoding strategies employed in this study and in nonhuman primates may be part of a solution for humans. Future experiments will have to evaluate the viability of a combined strategy.

REVIEWER's COMMENT: pg. 3, line 28. Define the parenthetical quantification of the swing phase, i.e., "18.2 ± 5.3 %" of what.

AUTHORS' ANSWERS: Refers to the % of swing phase

ACTION IN THE MANUSCRIPT: Now reads (**dragging: 18.2 ± 5.3 % of swing phase duration**).

REVIEWER's COMMENT: pg. 4, 3rd paragraph. It is not clear how accurate the thresholded normalized cumulative firing predicted the "100 ms anticipation" of foot-off times. Data are presented on the success of the decoder in predicting foot-off times, but not how well it predicts the time 100 ms before foot-off. In the next paragraph, the average time of stimulation onset relative to foot-off is reported, but it would be interesting to know the distribution of predicted 100 ms anticipation times without stimulation. This will indicate how well a single threshold predicted the 100 ms anticipation events from step to step.

AUTHORS' ANSWERS: In the absence of stimulation, the prediction anticipated foot off-events by 28.9 ± 14.8 ms on average. Instead, by simply lowering the detection threshold, we programmed the delivery of the stimulation 100 ms in advance to allow the burst to directly trigger a powerful swing phase without dragging, thus exactly 102.2 ± 24.5 ms after the detection.

ACTION IN THE MANUSCRIPT: We have added this information in the Results:

During continuous stimulation, the decoder detected foot-off events 28.9 ± 14.8 ms prior to the onset of these events. When the brain-spine interface triggered the onset of a stimulation burst with 100 ms anticipation, the detection occurred, as predicted, 102.2 ± 24.5 ms before foot-off events..

REVIEWER's COMMENT 5: Fig. 3 legend. The legend for Step 4A states that stimuli were delivered when the normalized cumulative firing crossed a threshold corresponding to 200 ms before foot-off – the text in the Results and Methods sections states 100 m.

AUTHORS' ANSWERS: Many thanks for noting this mistake. The correct value is 100 ms.

ACTION IN THE MANUSCRIPT: We corrected this mistake.

REVIEWER's COMMENT: Figs. 3c, 7c,d. The shaded regions around the average traces are presumably a measure of variance, but this should be specified in the legends".

AUTHORS' ANSWERS: We have adjusted the legend of the figures as suggested.

REVIEWER's COMMENT: pg. 4, paragraph 4. Indicate the stimulus current that was used for the binary brain-spine interface.

AUTHORS' ANSWERS: This is a very important point that we had forgotten to emphasize. Due to the delivery of a short burst, the amplitude of the stimulation could be increased compared to continuous stimulation. Together with the timing of the burst, this larger amplitude of stimulation was essential in mediating the improvement of locomotor performance with the binary brain-spine interface.

ACTION IN THE MANUSCRIPT: We have stressed this point in the Results:

During continuous stimulation, the amplitude of the current delivered to the L1-L2 segment was optimal around 185 ± 38 μ A on average. The delivery of short stimulation bursts allowed to increase the stimulation amplitude, which reached 262 ± 117 μ A on average during locomotion with the binary brain-spine interface. The occurrence of well-timed bursts of higher stimulation amplitude improved locomotor performance.

REVIEWER's COMMENT: Fig. 4a shows an example similar to Fig. 2a and is probably not necessary.

AUTHORS' ANSWERS: We agree with the reviewer.

ACTION IN THE MANUSCRIPT: We have removed this panel from Figure 4.

REVIEWER's COMMENT: Fig. 5a legend. How was it determined that the evoked potentials in the shaded area were due to direct stimulation of the motor nerve?

AUTHORS' ANSWERS: We and other groups (Edgerton, Gerasimenko, Minassian) have conducted extensive electrophysiological and pharmacological experiments to identify the origin of the motor responses elicited by EES. In these studies, we showed that EES elicited both short-latency direct responses due to the direct activation of the efferent nerve, and trans-synaptic responses to the activation of proprioceptive afferents. The direct responses occurred with a very short latency of 2 to 3

ms which corresponds to the time required for the elicited volley to travel from the location of the stimulation on the efferent nerve to the muscle. Instead, trans-synaptic responses involve the travel of evoked afferent volley to the spinal cord (~0.5ms), at least one synaptic event (~2.5ms), and the travel of the efferent volley to the muscle (~2.5 ms). We thus used the sudden occurrence of a short latency responses when increasing the stimulation amplitude to identify the onset of the undesired direct responses (see Capogrosso et al., J Neuroscience 2012 or Martin et al. Neuron 2016 for more details).

ACTION IN THE MANUSCRIPT: We have added this information in the Legend:

The shaded areas distinguish direct responses (direct stimulation of the motor nerve) from post-synaptic responses, which are elicited from the recruitment of proprioceptive feedback circuits. These temporal windows are defined from the expected latencies and durations of these responses.

REVIEWER's COMMENT: Fig. 5b legend. Specify the measure of response amplitudes that was used.

AUTHORS' ANSWERS: We measured the amplitude of the responses from the average EMG envelope.

ACTION IN THE MANUSCRIPT: We have added this information in the Legend:

The amplitude of motor responses was calculated as the integral of the averaged and rectified signals over the temporal window compatible with trans-synaptic responses (n = 10 repetitions per amplitude).

REVIEWER's COMMENT: pg. 4, last paragraph. Indicate whether these correlations were calculated with no stimulation, continuous spinal stimulation, or the binary-spine interface.

AUTHORS' ANSWERS: The correlations were calculated during continuous stimulation, as indicated in the legend of the related figure.

REVIEWER's COMMENT: pg. 5, 2nd to last paragraph. The term "step height" should be used rather than "locomotor performance" since it is the only parameter shown in Fig. 6B.

AUTHORS' ANSWERS: We have completely rewritten this section and added new results, as explained above.

REVIEWER's COMMENT: pg. 5, last paragraph. Indicate the time after injury that the stair climbing task was performed.

AUTHORS' ANSWERS: We have added this information in the new section on the staircase.

ACTION IN THE MANUSCRIPT: Results:

For this purpose, we tested the rats one month after injury during a stair climbing task that required a voluntary increase in foot clearance during swing (Fig. 7A).

REVIEWER's COMMENT: Fig. 7b legend. Define a "tumble."

ACTION IN THE MANUSCRIPT: This figure is now figure 8. In this figure, we colored the stick diagram in specific colors to highlight the definition of a fall and tumble. We also defined these terms in the main text and in the Legend:

Circular plots reporting the relative percent of trials with a successful step onto the elevated platform (pass), a tumble (hitting the foot against the staircase) and a fall when climbing the staircase with continuous stimulation or proportional brain-spine interface (n = 5 rats).

REVIEWER's COMMENT: pg.6, section on rehabilitation enabled by the proportional brain-spine interface. Indicate whether the animals were trained on the treadmill or overground.

AUTHORS' ANSWERS: They were trained in both conditions in the same amount.

ACTION IN THE MANUSCRIPT: Results:

Both groups of rats were trained for approximately 10 min on the treadmill before performing locomotion overground and along a staircase during 20 min.

REVIEWER's COMMENT: pg. 12, line 15. The Precision Systems and Instrumentation device is called the "Infinite Horizon Impactor."

AUTHORS' ANSWERS: Many thanks for noticing this mistake.

ACTION IN THE MANUSCRIPT: Corrected

REVIEWER's COMMENT: pg. 12, line 5 from the bottom. Please add another citation to the paper(s) that best describe(s) the correlation method.

ACTION IN THE MANUSCRIPT: We cited the following paper:

Centering, scaling, and transformations: improving the biological information content of metabolomics data Robert A van den Berg Email author, Huub CJ Hoefsloot, Johan A Westerhuis, Age K Smilde and Mariët J van der Werf BMC Genomics 20067:142

REVIEWER's COMMENT: pg. 13, section on cortical recordings. The description of the multi-unit recordings seems unnecessarily complicated. I guess that a spike-time event was recorded whenever the signal rose through the threshold level. If so, it seems confusing to use the phrase "field potential stochastic events."

ACTION IN THE MANUSCRIPT: We modified the sentence as follows.

We calculated multi-unit activity (MUA) from **the neural signal** crossed a threshold value defined visually for each channel.

REVIEWER's COMMENT: pg. 13, section on cortical decoding. Please describe the method used to determine which 6 MUA channels "correlated most" with muscle activity.

AUTHORS' ANSWERS: We calculated these correlations using the Matlab function `corrcoeff`. Practically, we calculated the correlation between each channel MUA signal and the envelope of the EMG signal that was rectified and low-pass filtered at 10 Hz.

ACTION IN THE MANUSCRIPT: We explained this procedure in the Methods:

For each rat, we identified the 6 channels with the MUA that correlated most with the envelope of the tibialis anterior muscle. **The EMG signal was rectified and low-pass filtered at 10 Hz before applying the Matlab function `corrcoeff` between this signal and each neural signal.**

REVIEWER's COMMENT: pg. 13, section on cortical decoding. More information on the calculation of the normalized cumulative firing would be helpful. In the figures, the normalized cumulative firing appears to be a continuous variable, or at least has more amplitude and temporal resolution than could be obtained by binning 6 channels of spike events. Please clarify how the channels were normalized, how they were combined, and how the weights were determined for the linear combination.

AUTHORS' ANSWERS: We agree that the description of this variable was not sufficiently detailed.

ACTION IN THE MANUSCRIPT: We have adapted the Methods section as follows:

The hard-real-time controller operated within cycles of 12 kHz. The *normalized cumulative firing* was thus a continuous signal generated with a frequency of 12 kHz. Its value instantly increased by a fixed equal quantity δ whenever a spike was detected in one of the six selected channels. The normalized cumulative firing decreased with a Gaussian decay over time (Finite Impulse Response filter with sample decay of 80% in 40 ms). The value δ was normalized in order to ensure that the peak normalized cumulative firing equals one over the duration (approximately 30s) of the data used for calibrating the decoder.

We also adjusted the new Figure 3 to explain better how the *normalized cumulative firing* was elaborated, since this novel variable was critical for the robustness and ease-of-use of the brain-spine interface for gait rehabilitation.

REVIEWER's COMMENT: pg. 13, section on binary-spine interface. The term "flexion detections" is not defined (I guess it's equivalent to "event detection" but this should be clear).

ACTION IN THE MANUSCRIPT: We modified the section as follows:

When evaluating decoder accuracy, all the **detected foot-off events** that occurred within a window of [-200, 100] ms centred on the actual foot-off (roughly 25% of the average step cycle) were considered as true positives (**foot-off detected**).

REVIEWER's COMMENT: pg. 14. section on analysis of spinal cord damage. Were the sections labeled with a cell body or myelin stain? The description of the measurement ("The amount of spared tissue was computed as the ratio between the number of pixels at the epicenter and in the intact sections.") is unclear; please provide additional information.

AUTHORS' ANSWERS: All the spinal cord sections were labeled against GFAP to visualize neural versus non-neural tissues. The epicenter of the spinal cord injury was imaged with a microscope, and a bitmap image was generated. We built a script that took this image and calculated the number of pixels from the spared tissue and from the lesion/background and calculated the ratio between both values.

ACTION IN THE MANUSCRIPT: We modified the text as follows:

Analysis of spinal cord damage. The extent and location of spinal cord damage was evaluated in each rat. **The damaged region of the spinal cord was cut in 40 µm thick coronal sections that were stained with an antibody against glial fibrillary acidic protein (GFAP). Briefly, spinal cord sections were incubated overnight in serum containing anti-GFAP (1:1000, Dako, USA) antibodies. Immunoreactions were visualized with secondary antibodies labelled with Alexa fluor® 555. Sections were mounted onto microscope slides using anti-fade fluorescent mounting medium and covered with a cover-glass. The sections corresponding to the lesion epicentre and to the first intact sections immediately rostral and caudal to the injury were selected for each rat, and then imaged using the Olympus Slide Scanner VS120–L100 microscope at 10x magnification. Custom-written Matlab scripts were used to analyse the image. Briefly, the images were divided into square regions of interest (ROI). Files were color-filtered and binarized by means of intensity thresholds that was set empirically and then maintained constant across all the sections. For each lesion core, the number of pixels contained within the regions with spared tissue was calculated. For each spinal cord, this value was compared to the average pixel size of the first intact sections located rostrally and caudally to the lesion. The ratio between both values was used as the amount of spared tissue.**

REVIEWER's COMMENT: Supplementary Fig. 3c,d. Can the mean values for the individual rats be plotted for the gait rehabilitation as for continuous stimulation and binary brain-spine interface?

ACTION IN THE MANUSCRIPT: The individual values were added to the figure.

REVIEWERS' COMMENTS:

Reviewer #1 (Remarks to the Author):

The authors have done a commendable job of addressing my major concerns. I have a few small points that should be addressed.

It is unclear why the foot-off event decoder performs differently for continuous stimulation (28.9 ± 14.8 ms prior) and brain-spine interface (102.2 ± 24.5 ms). It might be helpful if additional details or explanation of the decoder is provided. What is the history of neural data that the decoder uses?

Page-1 "all the technical and practical features" – this seems too comprehensive a claim

Page-5 Figure reference 3C and 3B are swapped

Page-16 "delivered continuous" should read as "delivered continuously"

Page-16 "Euclidian" should be "Euclidean"

Page-24 Figure 6a uses the acronym EES for the first time without prior expansion in the document. Are the pulses bipolar? Mention it somewhere, possibly with a figure inset.

Reviewer #2 (Remarks to the Author):

The resubmitted manuscript has been substantially improved by clarifying the experimental methods, providing additional results, and by more clearly describing how the results differ from the lab's previous studies. Most of the points from my original review have been adequately addressed, but I feel further comment is needed on two of the major points, despite the authors' in depth responses, which are greatly appreciated.

First, the authors agreed with both reviewers to reduce their discussion of neuroplasticity. Nonetheless, the second paragraph of the Introduction is devoted entirely (except for its last sentence) to plasticity. This seems out of place since the results do not bear directly on this issue. The authors raise the issue of neuroplasticity again in the Discussion, and this seems appropriate, since a discussion of the mechanisms underlying the recovery described in Results is relevant. However, I recommend that the paragraph on plasticity be deleted from the Introduction.

Second, it is still not clear to me that the paper's findings are sufficiently "novel" and "interesting to other related disciplines," and of "extreme importance to scientists in the specific field" - 3 of the 4 criteria required for a paper in a Nature Research journal (per the publisher's website) - to merit publication in Nature Communications. To address my concern that the advances of the current study were somewhat incremental, the authors have added a new section in the Introduction in which they define three challenges that are tackled in the current study – a brain-spinal interface that does not require training prior to injury, that requires minimal time for calibration, and that enables training in natural tasks such as overground walking. However, each of these challenges was already accomplished in the lab's earlier study (Capogrosso et al) in non-human primates using brain-controlled, binary (i.e., on-off), spinal stimulation to improve treadmill and overground locomotion after

spinal cord injury. The current study demonstrates that proportional stimulation gives a higher level of control that produces further improvements in locomotor performance, including on a more challenging stair climbing task. But similar brain-controlled proportional stimulation protocols have been used in earlier studies in animals and humans (if not in the exact same way as in the current application). To my mind, the most significant advance is the finding that the proportional interface results in a greater, or at least earlier post-injury (it's not entirely clear which), measure of volitional control of cortical activity to drive the interface. Although very interesting, even this finding doesn't seem to be a significant advance in terms of translational potential – it is well known that primates can volitionally modulate brain activity to control a brain-computer interface. Therefore, I remain unconvinced that the current results “represent an advance in understanding likely to influence thinking in the field” (as summarized in the Nature criteria for publication).

Minor points

1. It is not clear how the swing phase is terminated with the proportional interface. If modulated cortical activity is at least partially (completely for several days after injury?) driven by proprioceptive signals related to foot height, and stimulation increases flexor muscle activity (which would further increase foot height), it would seem that a positive feedback loop would be established. The figures show clearly that the cumulative firing parameter decreases as foot height continues to increase during the swing phase. Can the authors explain how the swing phase is terminated with the proportional stimulation, especially before cortical activity leads leg flexion in the early phases of training?
2. It is not entirely clear from the figures if there is a threshold value for the cumulative firing parameter above which the stimulation is turned on during proportional control. It would be helpful if this were explicitly stated in the Methods section.
3. pg. 7, first 2 paragraphs and Fig. 6. The figure shows clearly that the increase of the cumulative firing parameter occurs before the increase in foot height, in contrast to cortical activity lagging the motor output with the binary interface (Fig. 3). This is a highly significant finding. Although it is described in the Discussion section, it is worth highlighting this result here.
4. pg. 7, last paragraph, and Fig. 7. Please describe the measure that was used to quantify “intra-limb coordination.”
5. pg. 16, second paragraph. “Infinite Horizon” should be capitalized.
6. pg. 17, section on cortical decoding. The description of the normalized cumulative firing parameter is still confusing. What are the 12 kHz cycles (spike timing resolution was 83 microseconds)? What was the calibration procedure for the decoder (i.e., what was the rat doing during the ~30 s of cortical recording)? Fig 3a gives an equation for cumulative firing that includes weighting parameters, but the description in Methods does not mention any weighting. Please clarify.
7. pg. 17, section of binary brain-spinal interface. The sentence “True negatives...” is

repeated.

8. Figure 3 legend. I think it would be clearer if the description of Step 4 indicated that the stimulation was triggered based on the "predicted" occurrence of foot-off events: "When the cumulative firing crossed a threshold corresponding to 100 ms before the **predicted** occurrence of foot-off events, the pulse generator...."

9. Fig. 8 legend. The legend refers to a bar plot in panel d, but there is no bar plot in d.

10. Fig. 9 legend. In panel a, the confusion matrix for decoding from ankle flexor is on top and for brain activity on the bottom. The legend refers to them in the reverse order. Are either the labels or the confusion matrices switched?

11. In some places the English is a bit rough. The paper would benefit from an English editor tightening the language

NCOMMS-17-29883 R2

BRAIN-CONTROLLED MODULATION OF SPINAL CIRCUITS IMPROVES RECOVERY FROM SPINAL CORD INJURY

Reviewer #1 (Remarks to the Author):

The authors have done a commendable job of addressing my major concerns. I have a few small points that should be addressed.

AUTHORS' ANSWERS: Many thanks for your appreciation of our work

It is unclear why the foot-off event decoder performs differently for continuous stimulation (28.9 ± 14.8 ms prior) and brain-spine interface (102.2 ± 24.5 ms).

AUTHORS' ANSWERS: These two experiments were performed with different goals. For continuous stimulation, the aim was to detect foot-off with the highest degree of precision, which we achieved with a 29 ms accuracy (anticipation). For the experiments with the BSI, the aim was to anticipate the event in order to have sufficient time to deliver a burst of stimulation over the spinal cord in order to enhance flexion. We thus tuned the decoder to anticipate the detection by 100ms, and obtained an average of 102 ms.

ACTION: We added one sentence to clarify:

“The decoder anticipated foot-off events by 100 ms, which enabled the delivery of the stimulation at the relevant timing to promote flexion.”

It might be helpful if additional details or explanation of the decoder is provided. What is the history of neural data that the decoder uses?

AUTHORS' ANSWERS: The methods section “Cortical decoding” provide this information. Any spike encountered adds a value δ to the normalized cumulative firing, which is the history of neural data. The history decays Gaussianly over time (FIR with sample decay of 80% in 40 ms). Additional information that we could add is that the FIR history length is 80ms (that is 100% decay). To account for the various delays, we tuned the decoder to anticipate foot-off events by 100 ms.

Page-1 “all the technical and practical features” – this seems too comprehensive a claim

AUTHORS' ANSWERS: We agree, the “all” is in excess

ACTION: “all” has been removed.

Page-5 Figure reference 3C and 3B are swapped

ACTION: Many thanks. Corrected.

Page-16 “delivered continuous” should read as “delivered continuously”

ACTION: Many thanks. Corrected.

Page-16 “Euclidian” should be “Euclidean”

ACTION: Many thanks. Corrected.

Page-24 Figure 6a uses the acronym EES for the first time without prior expansion in the document.

ACTION: We have removed the acronym.

Are the pulses bipolar? Mention it somewhere, possibly with a figure inset.

AUTHORS' ANSWERS: All the pulses are biphasic, charged-balance monopolar pulses.

ACTION: "Biphasic monopolar" is now indicated in the Methods section "Spinal cord stimulation".

Reviewer #2 (Remarks to the Author):

The resubmitted manuscript has been substantially improved by clarifying the experimental methods, providing additional results, and by more clearly describing how the results differ from the lab's previous studies. Most of the points from my original review have been adequately addressed, but I feel further comment is needed on two of the major points, despite the authors' in depth responses, which are greatly appreciated.

AUTHORS' ANSWERS: Many thanks for your appreciation of the effort invested to clarify the manuscript and respond to the various queries.

First, the authors agreed with both reviewers to reduce their discussion of neuroplasticity. Nonetheless, the second paragraph of the Introduction is devoted entirely (except for its last sentence) to plasticity. This seems out of place since the results do not bear directly on this issue. The authors raise the issue of neuroplasticity again in the Discussion, and this seems appropriate, since a discussion of the mechanisms underlying the recovery described in Results is relevant. However, I recommend that the paragraph on plasticity be deleted from the Introduction.

AUTHORS' ANSWERS: We have taken this recommendation into consideration, and modified the text accordingly:

ACTION: The paragraph has been removed, and replaced by a short statement on our hypothesis:

"While these neural bypasses aim at restoring lost motor functions, there is mounting evidence that their long-term use during rehabilitation may augment functional recovery⁶⁻⁸. Our objective was to evaluate this possibility. Specifically, we previously showed^{9,10} that gravity-assisted gait rehabilitation enabled by continuous electrical spinal cord stimulation restores voluntary control of locomotion after a severe spinal cord injury (SCI) leading to paralysis. Here, we hypothesized that, compared to continuous stimulation, a direct cortical control over adaptive stimulation protocols during rehabilitation would enhance this locomotor recovery. "

Minor points

1. It is not clear how the swing phase is terminated with the proportional interface. If modulated cortical activity is at least partially (completely for several days after injury?) driven by proprioceptive signals related to foot height, and stimulation increases flexor muscle activity (which would further increase foot height), it would seem that a positive feedback loop would be established. The figures show clearly that the cumulative firing parameter decreases as foot height continues to increase during the swing phase. Can the authors explain how the swing phase is terminated with the proportional stimulation, especially before cortical activity leads leg flexion in the early phases of training?

AUTHORS' ANSWERS: There is a number of potential mechanisms that may account for the termination of the swing phase despite the presence of the stimulation that could theoretically increase cortical activity. For example, there is a robust inhibition of afferent activity towards the end of the swing phase. This inhibition enables epidural electrical stimulation to be directed towards extensor motoneurons, and thus to promote alternation between extension and flexion. We have documented this mechanism in great details in a previous publication (Martin et al., Neuron 2016). Another possibility is the intrinsic modulation of the motor cortex during locomotion, which would naturally trigger the decrease in stimulation, as observed in the present study. Indeed, motor cortex activity peaks during early swing and then declines, both in healthy animals and in our model of SCI. This modulation appears preserved while the rats operate the proportional BSI, despite the presence of the stimulation, suggesting that the mechanisms responsible for motor cortex modulation remains functional. At this stage, we feel that this consideration are too speculative. One certitude is the adequate decline in motor cortex activity after early swing, which enables the smooth operations of the BSI. For now, we would prefer avoiding engaging into speculative considerations.

2. It is not entirely clear from the figures if there is a threshold value for the cumulative firing parameter above which the stimulation is turned on during proportional control. It would be helpful if this were explicitly stated in the Methods section.

AUTHORS' ANSWERS: This point is correct. The Normalized firing means peaks at 1, which is reported in the Methods. However, this value also goes down to 0, which was not mentioned. Troughs at 0 are obtained by adding a constant negative bias to the equation in Figure 3, step 3 and Figure 6, step 3. When occasionally during operation the normalized firing goes below zero, EES amplitude is saturated to the minimum functional value. This is equivalent to the opposite case: if normalized firing goes above 1, EES is saturated to the maximum functional amplitude

ACTION: We have added the bias b in the equations and updated the figures accordingly.

3. pg. 7, first 2 paragraphs and Fig. 6. The figure shows clearly that the increase of the cumulative firing parameter occurs before the increase in foot height, in contrast to cortical activity lagging the motor output with the binary interface (Fig. 3). This is a highly significant finding. Although it is described in the Discussion section, it is worth highlighting this result here.

AUTHORS' ANSWERS: We agree with the importance of this finding.

ACTION: We have added a statement in the conclusions of the related Results section:

"These results suggest that rats were able to anticipate the modulation of motor cortex activity to mediate a functional increase in leg flexion."

4. pg. 7, last paragraph, and Fig. 7. Please describe the measure that was used to quantify "intra-limb coordination."

AUTHORS' ANSWERS: These values were obtained by calculating the cross-correlation between time series of the elevation angles of adjacent segments.

ACTION: Reported in the Results:

(cross-correlation between leg-foot $P < 0.001$, thigh-leg $P=0.003$, crest-thigh $P = 0.3$, t-test, **Fig. 7E**).

5. pg. 16, second paragraph. "Infinite Horizon" should be capitalized.

ACTION: Many thanks. Corrected.

6. pg. 17, section on cortical decoding. The description of the normalized cumulative firing parameter is still confusing. What are the 12 kHz cycles (spike timing resolution was 83 microseconds)? What was the calibration procedure for the decoder (i.e., what was the rat doing during the ~30 s of cortical recording)? Fig 3a gives an equation for cumulative firing that includes weighting parameters, but the description in Methods does not mention any weighting. Please clarify.

AUTHORS' ANSWERS: The vector w is composed of 32 values. Only 6 values are non-zero. This means that 6 channels over 32 are selected. These six entries have the value δ

ACTION: These precisions are reported in the section Methods / Cortical decoding.

7. pg. 17, section of binary brain-spinal interface. The sentence "True negatives..." is repeated.

ACTION: Many thanks. Corrected.

8. Figure 3 legend. I think it would be clearer if the description of Step 4 indicated that the stimulation was triggered based on the "predicted" occurrence of foot-off events: "When the cumulative firing crossed a threshold corresponding to 100 ms before the **predicted** occurrence of foot-off events, the pulse generator..."

ACTION: Many thanks. Corrected as suggested.

9. Fig. 8 legend. The legend refers to a bar plot in panel d, but there is no bar plot in d.

AUTHORS' ANSWERS: The new figure only reports trajectory as a trace, as opposed to a bar plot as previously. The legend had not been corrected.

ACTION: Many thanks for noting this error.

10. Fig. 9 legend. In panel a, the confusion matrix for decoding from ankle flexor is on top and for brain activity on the bottom. The legend refers to them in the reverse order. Are either the labels or the confusion matrices switched?

ACTION: Many thanks. Corrected.

11. In some places the English is a bit rough. The paper would benefit from an English editor tightening the language

ACTION: We have edited the English in some places.